

Present and future aerosol impacts on Arctic climate change in the GISS-E2.1 Earth system
model
Ulas Im[1,2,*], Kostas Tsigaridis[3,4], Gregory Faluvegi[3,4], Peter L. Langen[1,2], Joshua P. French[5],
Rashed Mahmood[6], Thomas Manu[7], Knut von Salzen[8], Daniel C. Thomas[1,2], Cynthia H.
Whaley[8], Zbigniew Klimont[9], Henrik Skov[1,2], Jørgen Brandt[1,2]
[1] Department of Environmental Science, Aarhus University, Roskilde, Denmark.
[2] Interdisciplinary Centre for Climate Change, Aarhus University, Roskilde, Denmark.
[3] Center for Climate Systems Research, Columbia University, New York, NY, USA.
[4] NASA Goddard Institute for Space Studies, New York, NY, USA.
[5] Department of Mathematical and Statistical Sciences, University of Colorado Denver, USA.
[6] Barcelona Supercomputing Center, Barcelona, Spain.
[7] Swedish Meteorological and Hydrological Institute, Norrköping, Sweden.
[8] Candian Centre for Climate Modelling and Analysis, Environment and Climate Change Canada,
Victoria, British Columbia, Canada.
[9] International Institute for Applied Systems Analysis (IIASA), Laxenburg, Austria.
[*] Corresponding author
Abstract
The Arctic is warming two to three times faster than the global average, partly due to changes
in short-lived climate forcers (SLCFs) including aerosols. In order to study the effects of
atmospheric aerosols in this warming, recent past (1990-2014) and future (2015-2050)
simulations have been carried out using the GISS-E2.1 Earth system model to study the
aerosol burdens and their radiative and climate impacts over the Arctic (>60 °N), using
anthropogenic emissions from the Eclipse V6b and the Coupled Model Intercomparison
Project Phase 6 (CMIP6) databases.
Surface aerosol levels, in particular black carbon (BC) and sulfate ($SO_4^{2-}$), have been
significantly underestimated by more than 50%, with the smallest biases calculated for the
nudged atmosphere-only simulations. CMIP6 simulations performed slightly better in
simulating both surface concentrations of aerosols and climate parameters, compared to the
Eclipse simulations. In addition, fully-coupled simulations had slightly smaller biases in
aerosol levels compared to atmosphere only simulations without nudging.
Arctic BC, organic carbon (OC) and $SO_4^{2-}$ burdens decrease significantly in all simulations
following the emission projections, with the CMIP6 ensemble showing larger reductions in
Arctic aerosol burdens compared to the Eclipse ensemble. For the 2030-2050 period, both the
Eclipse Current Legislation (CLE) and the Maximum Feasible Reduction (MFR) ensembles
simulated an aerosol top of the atmosphere (TOA) forcing of -0.39±0.01 W m$^{-2}$, of which -
0.24±0.01 W m$^{-2}$ were attributed to the anthropogenic aerosols. The CMIP6 SSP3-7.0
scenario simulated a TOA aerosol forcing of -0.35 W m$^{-2}$ for the same period, while SSP1-
2.6 and SSP2-4.5 scenarios simulated a slightly more negative TOA forcing (-0.40 W m$^{-2}$), of
which the anthropogenic aerosols accounted for -0.26 W m$^{-2}$. Finally, all simulations showed


an increase in the Arctic surface air temperatures both throughout the simulation period. In
2050, surface air temperatures are projected to increase by 2.4 °C to 2.6 °C in the Eclipse
ensemble and 1.9 °C to 2.6 °C in the CMIP6 ensemble, compared to the 1990-2010 mean.
Overall, results show that even the scenarios with largest emission reductions lead to similar
impact on the future Arctic surface air temperatures compared to scenarios with smaller
emission reductions, while scenarios no or little mitigation leads to much larger sea-ice loss,
implying that even though the magnitude of aerosol reductions lead to similar responses in
surface air temperatures, high mitigation of aerosols are still necessary to limit sea-ice loss.

1.      Introduction
The Arctic is warming two to three times faster than the global average (IPCC, 2013;
Lenssen et al., 2019). This is partly due to internal Arctic feedback mechanisms, such as the
snow and sea-ice-albedo feedback, where melting ice leads to increased absorption of solar
radiation, which further enhances warming in the Arctic (Serreze and Francis, 2006).
However, Arctic temperatures are also affected by interactions with warming at lower
latitudes (e.g., Stuecker et al., 2018; Graversen and Langen, 2019; Semmler et al., 2020) and
by local in situ response to radiative forcing due to changes in greenhouse gases and aerosols
in the area (Shindell, 2007; Stuecker et al., 2018). In addition to warming induced by
increases in global atmospheric carbon dioxide ($CO_2$) concentrations, changes in short-lived
climate forcers (SLCFs) such as tropospheric ozone ($O_3$), methane ($CH_4$) and aerosols (e.g.
black carbon (BC) and sulfate ($SO_4^{2-}$)) in the Northern Hemisphere (NH), have contributed
substantially to the Arctic warming since 1890 (Shindell and Faluvegi, 2009; Ren et al.,
2020). This contribution from SLCFs to Arctic heating together with efficient local
amplification mechanisms put a high priority on understanding the sources and sinks of
SLCFs at high latitudes and their corresponding climatic effects.
SLCFs include all atmospheric species, which have short residence times in the atmosphere
relative to long-lived greenhouse gases and have the potential to affect Earth's radiative
energy budget. Aerosols are important SLCFs and are a predominant component of air
quality that affects human health (Burnett et al., 2018, Lelieveld et al., 2019). They mostly
affect climate by altering the amount of solar energy absorbed by Earth and are efficiently
removed from the troposphere within several days to weeks. Black carbon (BC), which is a
product of incomplete combustion and open biomass/biofuel burning (Bond et al., 2004:
2013), absorbs a high proportion of incident solar radiation and therefore warms the climate
system (Jacobson, 2001). Sulphate ($SO_4^{2-}$), which is formed primarily through oxidation of
sulphur dioxide ($SO_2$), absorbs negligible solar radiation and cools climate by scattering solar
radiation back to space. Organic carbon (OC), which is co-emitted with BC during
combustion, both scatters and absorbs solar radiation and therefore causes cooling in some
environments and warming in others. Highly reflective regions such as the Arctic are more
likely to experience warming effects from these aerosols (e.g., Myhre et al, 2013).



Aerosols also influence climate via indirect mechanisms. After depositing onto snow and ice
surfaces, BC can amplify ice melt by lowering the albedo and increasing solar heating of the
surface (AMAP, 2015). Aerosols also affect cloud properties, including their droplet size,
lifetime, and vertical extent, thereby influencing both the shortwave cooling and longwave
warming effects of clouds. Globally, this indirect cloud forcing from aerosols is likely larger
than their direct forcing, although the indirect effects are more uncertain and difficult to
accurately quantify (IPCC, 2013). Moreover, Arctic cloud impacts are distinct from global
impacts, owing to the extreme seasonality of solar radiation in the Arctic, unique
characteristics of Arctic clouds (e.g., high frequency of mixed-phase occurrence), and rapidly
evolving sea-ice distributions. Together, they lead to complicated and unique phenomena that
govern Arctic aerosol abundances and climate impacts (e.g., Willis et al., 2018; Abbatt et al.,
2019). The changes taking place in the Arctic have consequences for how SLCFs affect the
region. For example, reductions in sea-ice extent, thawing of permafrost, and humidification
of the Arctic troposphere can affect the emissions, lifetime and radiative forcing of SLCFs
within the Arctic (Thomas et al., 2019).

The effect of aerosols on the Arctic climate through the effects of scattering and absorption of
radiation, clouds, and surface ice/snow albedo has been investigated in previous studies (i.e.
Clarke and Noone, 1985; Flanner et al., 2007; Shindell et al., 2012; Bond et al., 2013;
Dumont et al., 2014). Arctic climate change through aerosols is mainly driven by a response
to remote forcings (Gagné et al., 2015; Sand et al., 2015; Westervelt et al., 2015). Lewinschal
et al. (2019) estimated an Arctic temperature change per unit sulfur emission of -0.020 to -
0.025 K per TgS yr$^{-1}$. Sand et al. (2020) calculated an Arctic surface air temperature response
of 0.06 - 0.1 K per Tg BC yr$^{-1}$ to BC emissions in Europe and North America, and slightly
lower response (0.05-0.08 K per Tg BC yr$^{-1}$) to Asian emissions. Breider et al. (2017)
reported a short-wave (SW) aerosol radiative forcing (ARF) of $-0.19 \pm 0.05$ W m$^{-2}$ at the top
of the atmosphere (TOA) over the Arctic, which reflects the balance between sulphate
cooling ($-0.60$ W m$^{-2}$) and black carbon (BC) warming ($+0.44$ W m$^{-2}$). Schacht et al. (2019)
calculated a direct radiative forcing of up to 0.4 W m$^{-2}$ over the Arctic using the ECHAM6.3-
HAM2.3 global aerosol-climate model. Markowicz et al. (2021), using the NAAPS radiative
transfer model, calculated the total aerosol forcing over the Arctic (>70.5 °N) of -0.4 W m$^{-2}$.
Ren et al. (2020) simulated 0.11 and 0.25 W m$^{-2}$ direct and indirect warning in 2014-2018
compared to 1980-1984 due to reductions in sulfate, using the CAM5-EAST global aerosol-
climate model. They also reported that the aerosols produced an Arctic surface warming of
+0.30 °C during 1980–2018, explaining about 20% of the observed Arctic warming observed
during the last four decades, while according to Shindell and Faluvegi (2009), aerosols
contributed 1.09 ± 0.81 °C to the observed Arctic surface air temperature increase of 1.48 ±
0.28 °C observed in 1976-2007. AMAP (2015), based on four ESMs, estimated a total Arctic
surface air temperature response due to the direct effect of current global combustion derived
BC, OC and sulfur emissions to be +0.35 °C, of which +0.40 °C was attributed to BC in the
atmosphere, +0.22 °C to BC in snow, -0.04 °C to OC and -0.23 °C to SO$_4^{2-}$. Samset et al.
(2018) showed that Arctic warming due to aerosol reductions can reach up to 4°C in some
locations, with a multi-model increase for the 60°N–90°N region being 2.8°C. In addition,
recent studies also suggest that as global emissions of anthropogenic aerosols decrease,





natural aerosol feedbacks may become increasingly important for Arctic climate (Boy et al.,
2019; Mahmood et al., 2019).

In this study, we carry out several simulations with the fully coupled NASA Goddard
Institute of Space Sciences (GISS) earth system model, GISS-E2.1 (Kelley et al., 2020) to
study the recent past and future burdens of aerosols as well as their impacts on TOA radiative
forcing and climate-relevant parameters such as surface air temperatures, sea-ice, and snow
over the Arctic (>60 °N). In addition, we investigate the impacts from two different emission
inventories; Eclipse V6b (Höglund-Isaksson et al.,2020; Klimont et al., 2021) vs. CMIP6
(Hoesly et al., 2018; van Marle et al., 2017: Feng et al.,2020), as well as differences between
atmosphere-only vs. fully-coupled simulations, on the evaluation of the model and the
climate impact. Section 2 introduces the GISS-E2.1 model, the anthropogenic emissions, and
the observation datasets used in model evaluation. Section 3 presents results from the model
evaluation as well as recent past and future trends in simulated aerosol burdens, radiative
forcing, and climate change over the Arctic. Section 4 summarizes the overall findings and
the conclusions.

2.      Materials and methods

2.1.      Model description

GISS-E2.1 is the CMIP6 version of the GISS modelE Earth system model, which has been
validated extensively over the globe (Kelly et al., 2020; Bauer et al., 2020) as well as
regionally for air pollutants (Turnock et al., 2020). A full description of GISS-E2.1 and
evaluation of its coupled climatology during the satellite era (1979–2014) and the recent past
ensemble simulation of the atmosphere and ocean component models (1850-2014) are
described in Kelly et al. (2020) and Miller et al. (2020), respectively. GISS-E2.1 has a
horizontal resolution of 2° in latitude by 2.5° in longitude and 40 vertical layers extending
from the surface to 0.1 hPa in the lower mesosphere. The tropospheric chemistry scheme
used in GISS-E2.1 (Shindell et al., 2013) includes inorganic chemistry of $O_x$, $NO_x$, $HO_x$, CO,
and organic chemistry of $CH_4$ and higher hydrocarbons using the CBM4 scheme (Gery et al.,
1989), and the stratospheric chemistry scheme (Shindell et al., 2013), which includes chlorine
and bromine chemistry together with polar stratospheric clouds.

In the present work, we used the One-Moment Aerosol scheme (OMA: Bauer et al., 2020 and
references therein), which is a mass-based scheme in which aerosols are assumed to remain
externally mixed and have a prescribed and constant size distribution, with the exception of
sea salt that has two distinct size classes, and dust that is described by a sectional model with
an option from 4 to 6 bins. The default dust configuration that is used in this work includes 5
bins, a clay and 4 silt ones, from submicron to 16 μm in size. The first three dust size bins can
be coated by sulfate and nitrate aerosols (Bauer & Koch, 2005). The scheme treats sulfate,
nitrate, ammonium, carbonaceous aerosols (black carbon and organic carbon, including the
$NO_x$-dependent formation of secondary organic aerosol (SOA) and methanesulfonic acid
formation), dust and sea-salt. The model includes secondary organic aerosol production, as



described by Tsigaridis and Kanakidou, (2007). OMA only includes the first indirect effect,
in which the aerosol number concentration that impacts clouds is obtained from the aerosol
mass as described in (Menon & Rotstayn, 2006). In addition to OMA, we have also
conducted a non-interactive tracers (NINT: Kelley et al., 2020) simulation from 1850 to
2014, with noninteractive (through monthly varying) fields of radiatively active components
(ozone and multiple aerosol species) read in from previously calculated offline fields from
the OMA version of the model, ran using the Atmospheric Model Intercomparison Project
(AMIP) configuration in Bauer et al. (2020) as described in Kelley et al. (2020). The NINT
model includes a tuned aerosol indirect effect following Hansen et al. (2005).
The natural emissions of sea salt, dimethylsulfide (DMS), isoprene and dust are calculated
interactively. Anthropogenic dust sources are not represented in GISS-E2.1. Dust emissions
vary spatially and temporally only with the evolution of climate variables like wind speed
and soil moisture (Miller et al., 2006). The AMIP type simulations (see section 2.3) uses
prescribed sea surface temperature (SST) and sea ice fraction during the recent past (Rayner
et al., 2003).
2.2.    Emissions
In this study, we have used two different emission datasets; the ECLIPSE V6b (Höglund-
Isaksson et al.,2020; Klimont et al., 2021), which has been developed with support of the EU-
funded Action on Black Carbon in the Arctic (EUA-BCA) and used in the framework of the
ongoing AMAP Assessment (AMAP, 2021), referred to as *Eclipse* in this paper, and the
CEDS emissions (Hoesly et al., 2018; Feng et al.,2020) combined with selected Shared
Socio-economic Pathways (SSP) scenarios used in the CMIP6 future projections (Eyring et
al., 2016), collectively referred to as *CMIP6* in this paper.
2.2.1. EclipseV6b emissions
The ECLIPSE V6b emissions dataset is a further evolution of the scenarios established in the
EU funded ECLIPSE project (Stohl et al., 2015; Klimont et al., 2017). It has been developed
with the global implementation of the GAINS (Greenhouse gas – Air pollution Interactions
and Synergies) model (Amann et al., 2011). The GAINS model includes all key air pollutants
and Kyoto greenhouse gases, where emissions are estimated for nearly 200 country-regions
and several hundred source-sectors representing anthropogenic emissions. For this work,
annual emissions were spatially distributed on 0.5°x0.5° lon-lat grids for nine sectors: energy,
industry, solvent use, transport, residential combustion, agriculture, open burning of
agricultural waste, waste treatment, gas flaring and venting, and international shipping. A
monthly pattern for each gridded layer was provided at a 0.5°x0.5° grid level. The ECLIPSE
V6b dataset, used in this study, includes an estimate for 1990 to 2015 using statistical data
and two scenarios extending to 2050 that rely on the same energy projections from the World
Energy Outlook 2018 (IEA, 2018) but have different assumptions about the implementation
of air pollution reduction technologies, as described below.


The Current Legislation (CLE) scenario assumes efficient implementation of the current air
pollution legislation committed before 2018, while the Maximum Feasible Reduction (MFR)
scenario assumes implementation of best available emission reduction technologies included
in the GAINS model. The technology implementation pace in the MFR scenario includes
constraints resulting from age structure and typical lifetime of technologies but no constraints
resulting from possible economic implications of required large investment in emission
reduction technology. The assumptions and the details for the CLE and MFR scenarios (as
well as other scenarios developed within the ECLIPSE V6b family) can be found in
Höglund-Isaksson et al. (2020) and Klimont et al. (in preparation).
The MFR scenario demonstrates the additional reduction potential of $SO_2$ emissions by up to
60% and 40%, by 2030 for Arctic Council member and observer countries respectively, with
implementation of best available technologies mostly in the energy and industrial sectors and
to a smaller extent via measures in the residential sector. The Arctic Council member
countries' maximum reduction potential could be fully realized by 2030 whereas in the
observer countries additional reductions of 15% to 20% would remain to be achieved
between 2030 and 2050.
2.2.2. CMIP6 emissions
The CMIP6 emission datasets include a historical time series generated by the Community
Emissions Data System (CEDS) for anthropogenic emissions (Hoesly et al., 2018; Feng et al.,
2020), open biomass burning emissions (van Marle et al., 2010 ), and the future emission
scenarios driven by the assumptions embedded in the Shared Socioeconomic Pathways
(SSPs) and Representative Concentration Pathways (RCPs) (Riahi et al., 2017) that include
specific air pollution storylines (Rao et al., 2017 ). Gridded CMIP6 emissions are aggregated
to nine sectors: agriculture, energy, industrial, transportation, residential–commercial–other,
solvents, waste, international shipping, and aircraft. SSP data for future emissions from
integrated assessment models (IAMs) are first harmonized to a common 2015 base-year
value by the native model per region and sector. This harmonization process adjusts the
native model data to match the 2015 starting year values with a smooth transition forward in
time, generally converging to native model results (Gidden et al., 2018). The production of
the harmonized future emissions data is described in Gidden et al. (2019).
2.2.3. Implementation of the emissions in the GISS-E2.1
In the GISS-E2.1 Eclipse simulations, the non-methane volatile organic carbons (NMVOC)
emissions are chemically speciated assuming the SSP2-4.5 VOC composition profiles. The
CMIP6 emissions have been pre-processed to include the agricultural waste burning
emissions from the EclipseV6b dataset, while the rest of the biomass burning emissions are
taken from the CMIP6 emissions. In addition to the biomass burning emissions, the aircraft
emissions are also taken from the CMIP6 database to be used in the Eclipse simulations. As
seen in Figure 1, the emissions are consistently higher in the CMIP6 compared to the Eclipse
emissions. The main differences in the two datasets are mainly over south-east Asia (not
shown) . The CMIP6 emissions are also consistently higher on a sectoral basis compared to



the Eclipse emissions. The figure shows that for air pollutant emissions, the CMIP6 SSP1-2.6
scenario and the Eclipse MFR scenario follow each other closely, while the Eclipse CLE
scenario is comparable with the CMIP6 SSP2-4.5 scenario for most pollutants; that is to some
extent owing to the fact that the $CO_2$ trajectory of the Eclipse CLE and the SSP2-4.5 are very
similar (not shown). A more detailed discussion of differences between historical Eclipse and
CMIP6 as well as CMIP6 scenarios are provided in Klimont et al. (in preparation).
2.3.    Simulations

In order to contribute to the AMAP Assessment report (AMAP, 2021), the GISS-E2.1 model
participated with AMIP-type simulations, which aim to assess the trends of Arctic air
pollution and climate change in the recent past, as well as with fully-coupled climate
simulations. Five fully-coupled Earth system models (ESMs) simulated the future (2015-
2050) changes of atmospheric composition and climate in the Arctic (>60°N), as well as over
the globe. We have carried out two AMIP-type simulations, one with winds nudged to NCEP
(standard AMIP-type simulation in AMAP) and one with freely varying winds, where both
simulations used prescribed SSTs and sea-ice (Table 1). In the fully-coupled simulations, we
carried out two sets of simulations, each with three ensemble members, that used the CLE
and MFR emission scenarios. Each simulation in these two sets of scenarios were initialized
from a set of three fully-coupled ensemble recent past simulations (1990-2014) to ensure a
smooth continuation from CMIP6 to Eclipse emissions.

In addition to the AMAP simulations, we have also conducted CMIP6-type simulations in
order to compare the climate aerosol burdens and their impacts on radiative forcing and
climate impacts with those from the AMAP simulations. As seen in Table 1, we have
conducted one transient fully-coupled simulation from 1850 to 2014, and a number of future
scenarios.

2.4.    Observations

The GISS-E2.1 ensemble has been evaluated against surface observations of BC, OC and
$SO_4^{2-}$, ground-based and satellite-derived AOD 550 nm, as well as surface and satellite
observations of surface air temperature, precipitation, sea surface temperature, sea-ice extent,
cloud fraction, and liquid and ice water content in 1995-2014 period. The surface monitoring
stations used to evaluate the simulated aerosol levels have been listed in Table S1 and S2 in
the supplementary materials.

*2.4.1. Aerosols*

Measurements of speciated particulate matter (PM), black carbon (BC), sulfate ($SO_4^{2-}$), and
organic carbon (OC) come from three major networks: the Interagency Monitoring of
Protected Visual Environments (IMPROVE) for the United States; the European Monitoring
and Evaluation Programme (EMEP) for Europe; and the Canadian Air Baseline
Measurements (CABM) for Canada (Table S1 and S2). In addition to these monitoring



networks, BC, OC, and $SO_4^{2-}$ measurements from individual Arctic stations were used in this
study. The individual Arctic stations are Fairbanks and Utqiagvik, Alaska (part of
IMPROVE, though their measurements were obtained from their PIs); Gruvebadet and
Zeppelin mountain (Ny Alesund), Norway; Villum Research Station, Greenland; and Alert,
Nunavut (the latter being an observatory in Global Atmospheric Watch-WMO, and a part of
CABM). The measurement techniques are briefly described in the supplement.
AOD at 500 nm from the AErosol RObotic NETwork (AERONET, Holben et al., 1998) was
interpolated to 550 nm AOD using the Ångström formula (Ångström, 1929). We also used a
new merged AOD product developed by Sogacheva et al. (2020) using AOD from different
satellite-based products. According to Sogacheva et al. (2020), this merged product could
provide a better representation of temporal and spatial distribution of AOD. However, it is
important to note that the monthly aggregates of observations for both AERONET and the
satellite products depend on availability of data and are not likely to be the true aggregate of
observations for a whole month when only few data points exist during the course of a
month. In addition, many polar orbiting satellites take one observation during any given day,
and typically at the same local time. Nevertheless, these data sets are key observations
currently available for evaluating model performances. Information about the uncertain
nature of AOD observations can be found in previous studies (e.g. Sayer et al., 2018; Sayer
and Knobelspiesse, 2019; Wei et al., 2019; Schutgens et al., 2020, Schutgens, 2020;
Sogacheva et al., 2020).
*2.4.2. Surface air temperature, precipitation, and sea-ice*
Surface air temperature and precipitation observations used in this study are from University
of Delaware gridded monthly mean data sets (UDel; Willmott and Matsuura, 2001). UDel's
0.5° resolution gridded data sets are based on interpolations from station-based measurements
obtained from various sources including the Global Historical Climate Network, the archive
of Legates and Willmott and others.  The Met Office Hadley Center's sea ice and sea surface
temperature (HadISST; Rayner et al., 2003) was used for evaluating model simulations of sea
ice and SSTs. HadISST data is an improved version of its predecessor known as global sea
ice and sea surface temperature (GISST). HadISST data is constructed using information
from a variety of data sources such as the Met Office Marine Database, Comprehensive
Ocean-Atmosphere Data Set, passive microwave remote sensing retrieval and sea ice charts.
*2.4.3. Satellite observations used for cloud fraction  and cloud liquid water and ice water*
The Advanced Very High Resolution Radiometer (AVHRR-2) sensors onboard the NOAA
and EUMETSAT polar orbiting satellites have been flying since the early 1980s. These data
have been instrumental in providing the scientific community with climate data records
spanning nearly four decades. Tremendous progress has been made in recent decades in
improving, training and evaluating the cloud property retrievals from these AVHRR sensors.
In this study, we use the retrievals of total cloud fraction from the second edition of
EUMETSATs Climate Monitoring Satellite Application Facility (CM SAF) Cloud, Albedo



and surface Radiation data set from AVHRR data (CLARA-A2, Karlsson et al., 2017). This
cloud property climate data record is available for the period 1982-2018. Its strengths and
weaknesses and inter-comparison with the other similar climate data records are documented
in Karlsson and Devasthale (2018). Further data set documentation including Algorithm
Theoretical Basis and Validation reports can be found in Karlsson et al. (2017).
Cloud liquid and ice water path estimates derived from the cloud profiling radar on board
CloudSat (Stephens et al., 2002) and constrained with another sensor onboard NASA's A-
Train constellation, MODIS-Aqua (Platnick et al., 2015), are used for the model evaluation.
These Level 2b retrievals, available through 2B-CWC-RVOD product (Version 5), for the
period 2007-2016 are analysed. This constrained version is used instead of its radar-only
counterpart, as it uses additional information about visible cloud optical depths from MODIS,
leading to better estimates of cloud liquid water paths. Because of this constraint the data are
available only for the day-lit conditions, and hence, are missing over the polar regions during
the respective winter seasons. The theoretical basis for these retrievals can be found in
http://www.cloudsat.cira.colostate.edu/sites/default/files/products/files/2B-CWC-
RVOD_PDICD.P1_R05.rev0_.pdf (last access: October 26th 2020). Being an active cloud
radar, CloudSat provides orbital curtains with a swath width of just about 1.4 km. Therefore,
the data are gridded at 5°x5° to avoid too many gaps or patchiness and to provide robust
statistics.
3.      Results
3.1.    Evaluation
The simulations are compared against surface measurements of BC, OC, $SO_4^{2-}$ and AOD, as
well as surface and satellite measurements of surface air temperature, precipitation, sea
surface temperature, sea-ice extent, total cloud fraction, liquid water path, and ice water path
described in section 2.4, by calculating the correlation coefficient ($r$) and normalized mean
bias ($NMB$).
*3.1.1. Aerosols*
The recent past simulations are for BC, OC, $SO_4$ and AOD (Table 2) against available surface
measurements, where individual time series for different stations are accumulated per species
in order to get an Arctic evaluation of the model. In addition to Table 2, the climatological
mean (1995-2014) of the observed and simulated monthly surface concentrations of  BC, OC,
$SO_4^{2-}$ and AOD at 550 nm (note that AOD is averaged over 2008, 2009 and 2014) are shown
in Figure 2. The AOD observation data for years 2008, 2009, and 2014 are used in order to
keep the comparisons in line with the multi-model evaluations being carried out in the
AMAP assessment report (AMAP, 2021). We also provide spatial distributions of the NMB,
calculated as the mean of all simulations for BC, OC, $SO_4$ and AOD in Figure 3. The
statistics for the individual stations are provided in the Supplementary Material, Tables S3-
S6. Results showed overall an underestimation of aerosol species over the Arctic, as
discussed below. Surface BC levels are underestimated at all Arctic stations from 15% to



90%. Surface OC levels are also underestimated from -5% to -70%, except for a slight
overestimation over Karvatn (<1%) and a large overestimation of 90% over Trapper Creek.
Surface $SO_4^{2-}$ concentrations are also consistently underestimated from -10% to -70%, except
for Villum Research Station over northeastern Greenland where there is an overestimation of
45%. Finally AODs are also underestimated over all stations from 20% to 60%. Such
underestimations at high latitudes have also been reported by many previous studies (e.g.
Skeike et al., 2011; Eckhardt et al., 2015; Lund et al., 2017, 2018; Schacht et al., 2019;
Turnock et al., 2020), pointing to a variety of reasons including uncertainties in emission
inventories, errors in the wet and dry deposition schemes, the absence or underrepresentation
of new aerosol formation processes, and the coarse resolution of global models leading to
errors in emissions and simulated meteorology. Turnock et al. (2020) evaluated the air
pollutant concentrations in the CMIP6 models, including the GISS-E2.1 ESM, and found that
observed surface $PM_{2.5}$ concentrations are consistently underestimated in CMIP6 models by
up to 10 μg m$^{-3}$, particularly for the Northern Hemisphere winter months, with the largest
model diversity near natural emission source regions and the Polar regions.
The BC levels are largely underestimated in simulations by 50% (CMIP6_Cpl_Hist) to 67%
(Eclipse_AMIP). The CMIP6 simulations have lower bias compared to EclipseV6b
simulations due to higher emissions in the CMIP6 emission inventory (Figure 1). Within the
EclipseV6b simulations, the lowest bias (-57%) is calculated for the Eclipse_AMIP_NCEP
simulation, while the free climate and coupled simulations showed a larger underestimation
(>62%), which can be attributed to a better simulation of transport to the Arctic when nudged
winds are used. The Eclipse simulations also show that the coupled simulations had slightly
smaller biases (*NMB*=-63%) compared to the AMIP-type free climate simulation  (AMIP-
OnlyAtm: *NMB*=-67%). The climatological monthly variation of the observed levels is
poorly reproduced by the model with *r* values around 0.3. BC levels are mainly
underestimated in winter and spring, while the summer levels are well captured by the
majority of the simulations (Figure 2).
Surface OC concentrations are underestimated from 8% (Eclipse_AMIP_NCEP) to 35%
(Eclipse_AMIP) by the Eclipse ensemble, while the CMIP6_Cpl_Hist simulation
overestimated surface OC by 13%. The Eclipse simulations suggest that the nudged winds
lead to a better representation of transport to the Arctic, while the coupled simulations had
smaller biases compared to the AMIP-type free climate simulation (AMIP-OnlyAtm), similar
to BC. The climatological monthly variation of the observed concentrations are reasonably
simulated, with *r* values between 0.51 and 0.69 (Table 2). The climatological monthly
variation of the OC levels are also well simulated in all seasons (Figure 2).
Surface $SO_4^{2-}$ levels are simulated with a smaller bias compared to the BC levels, however
still underestimated by 40% (CMIP6_Cpl_Hist) to 53% (Eclipse_AMIP_NCEP). The
Eclipse_AMIP_NCEP simulation is biased higher (NMB=-53%) compared to the
Eclipse_AMIP (NMB=-50%), probably due to higher cloud fraction simulated by the nudged
version (see section 3.1.6). The climatological monthly variation of observed $SO_4^{2-}$
concentrations are reasonably simulated in all simulations (r=0.65-0.74). The observed



springtime maximum is well captured by the GISS-E2.1 ensemble, with underestimations in
all seasons (Figure 2). The clear sky AOD over the Aeronet stations in the Arctic region is
underestimated by 33% (Eclipse_AMIP) to 47% (Eclipse_CplHist1). Similar negative biases
are found with comparison to the satellite based AOD product (Table 2). The climatological
monthly variation is poorly captured with *r* values between -0.07 to 0.07 compared to
AERONET AOD and 0 to 0.13 compared to satellite AOD. The simulations could not
represent the climatological monthly variation of the observed AERONET AODs (Figure 2).
*3.1.2. Climate*
The different simulations are evaluated against a set of climate variables and the statistics are
presented in Table 3 and in Figures 4 and 5. The climatological mean (1995-2014) monthly
Arctic surface air temperatures are slightly overestimated by up to 0.55 °C in the AMIP
simulations, while the coupled ocean simulations underestimate the surface air temperatures
by up to -0.17 °C. All simulations were able to reproduce the monthly climatological
variation with *r* values of 0.99 and higher (Figure 4). The monthly mean precipitation has
been underestimated by around 50% by all simulations (Table 3), with largest biases during
the summer and autumn (Figure 4). The observed monthly climatological mean variation was
very well simulated by all simulations, with *r* values between 0.80 and 0.90.
Arctic SSTs are largely underestimated by the ocean-coupled simulation up to -1.96 °C,
while the atmosphere-only runs underestimated SSTs by -1.5 °C (Table 3). The monthly
climatological mean variation is well captured with *r* values above 0.99 (Table 3, Figure 4),
with a similar cold bias in almost all seasons. The sea-ice extent was overestimated by all
coupled simulations by about 12%, while the AMIP-type Eclipse simulations slightly
underestimated the extent by 3% (Table 3). The observed variation was also very well
captured with very high *r* values. The winter and spring biases were slightly higher compared
to the summer and autumn biases (Figure 4).
All simulations overestimated the climatological (1995-2014) mean total cloud fraction by
21% to 25% during the extended winter months (October through February). The largest
biases were simulated by the atmosphere-only simulations, with the nudged simulation
having the largest bias (*NMB*=25%). The coupled model simulations are closer to the
observations during the recent past. On the other hand, the climatology of the cloud fraction
was best simulated by the nudged atmosphere-only simulation (Eclipse_AMIP_NCEP) with a
*r* value of 0.40, while other simulations showed a poor performance (*r*=-0.17 to +0.10),
except for the summer where the bias is lowest (Figure 5). The evaluation against CALIPSO
data however shows much smaller biases (*NMB* = +3% to +6%). This decrease in
overestimation is due to the strong underestimation of Arctic wintertime cloud formation by
AVHRR CLARA-A2 observations due to difficulties in separating cold and bright ice/snow
surfaces from clouds (Karlsson et al., 2017), leading to larger positive bias calculated for the
model.
Figure 5 shows the evaluation of the simulations with respect to LWP and IWP. It has to be
noted here that to obtain a better estimate of the cloud water content, the CloudSat



observations were constrained with MODIS observations which resulted in a lack of data
during the months with darkness (Oct-Mar) over the Arctic (see Section 2.4.3). Hence, we
present the results for the polar summer months only. As seen in Figure 5, all simulations
overestimated the climatological (2007-2014) mean Polar summer LWP by up to almost
75%. The smallest bias (14%) is calculated for the nudged atmosphere-only
(Eclipse_OnlyAtm_NCEP), while the coupled simulations had biases of 70% or more.
Observations show a gradual increase in the LWP, peaking in July, whereas the model
simulates a more constant amount for the nudged simulation and a slightly decreasing
tendency for the other configurations. All model simulations overestimate LWP during the
spring months. The atmosphere-only nudged simulations tend to better simulate the observed
LWP during the summer months (June through September). The coupled simulations,
irrespective of the emission dataset used, are closer to observations only during the months of
July and August.
The climatological (2007-2014) mean Polar summer IWP is slightly better simulated
compared to the LWP, with biases within -60% with the exception of the nudged Eclipse
(Eclipse_AMIP_NCEP) simulation (*NMB*=-74%). All simulations simulated the monthly
variation well, with *r* values of 0.95 and more.
In the Arctic, the net cloud forcing at the surface changes sign from positive to negative
during the polar summer (Kay and L'Ecuyer, 2013). This change typically occurs in May
driven mainly by shortwave cooling at the surface. Since the model simulates the magnitude
of the LWP reasonably, particularly in summer, the negative cloud forcing can also be
expected to be realistic in the model (e.g. Gryspeerdt et al. 2019). Furthermore, the aerosol
and pollution transport into the Arctic typically occurs in the lowermost troposphere where
liquid water clouds are prevalent during late spring and summer seasons. The interaction of
ice clouds with aerosols is, however, more complex, as ice clouds could have varying optical
thicknesses, with mainly thin cirrus in the upper troposphere and relatively thicker clouds in
the layers below. Without the knowledge on the vertical distribution of optical thickness, it is
difficult to infer the potential impact of the underestimation of IWP on total cloud forcing and
their implications.
3.2. Burdens
The recent past and future Arctic column burdens for BC, OC and $SO_4^{2-}$ for the different
scenarios and emissions are provided in Figure 6. The BC and $SO_4^{2-}$ burdens started
decreasing from the 1990s, while OC burden remains relatively constant, although there is
large year-to-year variability in all simulations. All figures show a decrease in burdens after
2015, except for the SSP3-7.0 scenario, where the burdens remain close to the 2015 levels.
The high variability in BC and OC burdens over the 2000's are due to the biomass burning
emissions from GFED, which have not been harmonised with the no-satellite era. It should
also be noted that these burdens can be underestimated considering the negative biases
calculated for the surface concentrations and in particular for the AODs reported in Table 2
and Tables S2-6.



All simulations show a significant negative BC burden trend (slope = -0.025±0.003 kTon yr$^{-1}$)
over the Arctic between 1990-2014, except for the CMIP6_Cpl_Hist, which shows a slight
non-significant increasing trend of 0.007 kTon yr$^{-1}$, which can be attributed to the large
increase in global anthropogenic BC emissions in CMIP6 after year 2000 (Figure 1). The
Eclipse ensemble also shows that the 1990-2010 mean BC burden is simulated to be similar
(3.4 kTon) in the coupled and AMIP-type simulations, while the nudged AMIP simulation
calculates a slightly higher burden (3.7 kTon). This can be attributed to a better resolved
transport of aerosols to the Arctic in the nudged simulation, as suggested by the model
evaluation (Table 2). From 2015 onwards, all simulations show a statistically significant
negative trend in the Arctic BC burden. The Eclipse simulations show a smaller negative in
the trend (-0.03±0.01 kTon yr$^{-1}$) compared to the CMIP6 simulations (-0.04±0.03 kTon yr$^{-1}$).
The Eclipse CLE simulations calculate a negative trend by -0.02±0.00 kTon yr$^{-1}$, leading to a
1.1 kTon decrease in the 2030-2050 mean compared to the 1990-2010 mean, while the
decrease is larger in the MFR scenario (-0.04±0.00 kTon yr$^{-1}$), leading to decrease of 2.3
kTon in of 2030-2050 mean. In the CMIP6 simulations, SSP1-2.6 gives the largest reduction
by -0.07 kTon yr$^{-1}$ (1.6 kTon decrease in 2030-2050 mean) while the smallest reduction is
simulated by the SSP3-7.0 simulation (-0.004 kTon yr$^{-1}$) with the 2030-2050 mean being 0.5
kTon lower than the 1990-2010 mean. The change in the Eclipse CLE scenario (-1.1 kTon) is
comparable with the change of -1 kTon in the SSP2-4.5 scenario, consistent with the
projected emission changes in the two scenarios (Figure 1).
The Eclipse simulations show overall a positive trend of OC between 1990 and 2014
(0.03±0.06 kTon yr$^{-1}$), however this trend is not significant at the 95% confidence level
(p=0.5-0.9). The CMIP6_Cpl_Hist simulation gives a larger trend (0.12 kTon yr$^{-1}$), similar to
the BC burden, due to a large increase in global anthropogenic OC emissions in CMIP6
(Figure 1). The nudged AMIP Eclipse simulation calculates the largest 1990-2010 mean OC
burden (57 kTon), while the coupled simulation shows a slightly lower 1990-2010 mean
burden (55 kTon). This largest OC burden in the Eclipse_AMIP_NCEP simulation is
attributed to the largest biogenic SOA calculated in this scenario (Figure S1). The
anthropogenic and biogenic contributions to SOA burdens in the coupled Eclipse and CMIP6
recent past simulations imply that the differences in the burdens between the two ensembles
can be attributed to the different anthropogenic emissions datasets used in the Eclipse and
CMIP6 simulations (Figure S1). The AMIP-type Eclipse run simulates a lower 1990-2010
mean OC burden (50 kTon), attributed to the smallest biogenic SOA burden in this scenario.
The Eclipse CLE simulations show a negative trend of -0.20±0.02 kTon yr$^{-1}$ between 2015
and 2050, leading to decrease of 6.2 kTon in 2030-2050 mean burden compared to the 1990-
2010 mean, while the MFR simulations show a steeper trend of -0.36±0.02 kTon yr$^{-1}$ (14.9
kTon decrease in 2030-2050 mean vs 1990-2010 mean). The CMIP6 simulations show a
much steeper trend of OC by -0.45±0.29 kTon yr$^{-1}$ compared to the Eclipse simulations, with
a decrease of 1.9 kTon (SSP3-7.0) to 17 kTon (SSP1-2.6) in the 2030-2050 mean compared
to the 1990-2010 mean. Similar to BC burdens, Eclipse CLE and CMIP6 SSP2-4.5 scenarios
project similar changes in 2030-2050 mean OC burden (6.9 kTon and 7.8 kTon,
respectively).




Regarding $SO_4^{2-}$ burdens, all simulations show a statistically significant negative trend both
in 1990-2014 and in 2015-2050, as seen in Figure 6. Eclipse and CMIP6 simulations show a
comparable decrease of Arctic sulfate burdens in the recent past period (-1.16±0.23 T yr$^{-1}$ and
-1.09 kTon yr$^{-1}$, respectively). Both the nudged AMIP-type and coupled simulations showed
a 1990-2010 mean $SO_4^{2-}$ burden of 92 kTon, while the AMIP-type simulation showed a
larger $SO_4^{2-}$ burden of 95 kTon, attributed to the larger cloud fraction simulated in this model
version (Table 2). The Eclipse CLE scenario shows a decrease of -0.14±0.02 kTon/yr in the
2015-2050 period, leading to a decrease of 28 kTon decrease in 2030-2050 mean compared
to the 1990-2010 mean, while the MFR shows a very similar trend of -0.15±0.03 kTon yr$^{-1}$,
however with a larger decrease of 2030-2050 mean (-38 kTon). On the other hand, the
CMIP6 simulation predicts a much larger decrease of sulfate burdens by -0.49±0.40 kTon yr$^{-1}$
in the future, largely driven by the SSP1-2.6 scenario that gives a decrease of -0.94 kTon yr$^{-1}$,
leading to a decrease of 45 kTon in 2030-2050 mean compared to the 1990-2010 mean.

3.3.    Arctic radiative forcing

The TOA aerosol radiative forcings over the Arctic as calculated by the sum of shortwave
and longwave TOA forcings from all aerosol species between 1850 and 2050 are presented in
Figure 7.  The instantaneous forcings are calculated with a double call to the model's
radiation code, with and without aerosols. The negative aerosol forcing has increased
significantly since 1850 until the 1970's due to an increase in aerosol concentrations. Due to
the efforts of mitigating air pollution and thus a decrease in emissions, the forcing became
less negative after the 1970's until 2015.

The coupled Eclipse simulations calculated an aerosol TOA radiative forcing of -0.32±0.01
W m$^{-2}$ for the 1990-2010 mean, while AMIP-type Eclipse simulations calculated a forcing of
-0.47 W m$^{-2}$ for the same period. For the 2030-2050 period, both the Eclipse CLE and MFR
ensembles simulated an aerosol TOA forcing of -0.39±0.01 W m$^{-2}$. For the anthropogenic
aerosols (Figure 7), the Eclipse TOA forcing in 1990-2010 is calculated to be -0.22±0.01 W
m$^{-2}$ by the Eclipse ensemble, while in the 2030-2050 period, the TOA anthropogenic forcing
(including biomass burning) became more negative in the Eclipse ensemble (-0.24±0.01 W
m$^{-2}$: -0.24±0.00 W m$^{-2}$ and -0.23±0.00 W m$^{-2}$ for CLE and MFR, respectively).

The forcing calculated for the individual aerosol species of BC, OC, $SO_4^{2-}$ and $NO_3^-$ are also
investigated separately (Table 4 and Figure 8). The increase in cooling effect of aerosols
calculated by the Eclipse ensemble is attributed mainly to the decrease in BC, which is
warming the atmosphere as opposed to other aerosol species (Figure 8). More negative
forcing is calculated for the OC and $NO_3^-$, while the $SO_4^{2-}$ forcing is becoming less negative
due to large reductions in $SO_2$ emissions (Figure 1). The net aerosol forcing is therefore
slightly more negative. In the CMIP6 ensemble, the BC forcing does not change as much
compared to the Eclipse ensemble to counteract the change in impact from $SO_4^{2-}$, giving a
slightly less negative net aerosol forcing. The CMIP6 ensemble also simulates a larger
increase in the negative $NO_3^-$ forcing compared to the Eclipse ensemble (Shindell et al.,



2013). Overall, the changes in the different aerosol species lead to a slightly different but less
negative net aerosol forcing by mid-century.
As seen in Table 4, the GISS-E2.1 ensemble calculated a BC TOA direct radiative forcing of
up to 0.23 W m$^{-2}$ over the Arctic, with both CMIP6 and Eclipse coupled simulations
estimating the highest forcing of 0.23 W m$^{-2}$ for the 1990-2010 mean. This agrees with
previous estimates of the BC direct forcing over the Arctic (e.g. Schacht et al., 2019). In the
future, the BC forcing is generally decreasing due to lower BC emissions, except for the
SSP3-7.0 scenario, where the BC forcing becomes more positive (0.28  W m$^{-2}$). The 1990-
2010 SO$_4^{2-}$ forcing is calculated to be up to -0.39 W m$^{-2}$ in the Eclipse simulations, while the
CMIP6 estimates a slightly more negative SO$_4^{2-}$ forcing (-0.4 W m$^{-2}$).  All future simulations
show a much less negative SO$_4^{2-}$ forcing in 2030-2050 due to the large reductions in SO$_2$
emissions. Both OC and NO$_3^-$ forcings are relatively smaller and negative compared to BC
and SO$_4^{2-}$ in the 1990-2010 period, and become more negative in 2030-2050.
The NINT and the CMIP6_Cpl_Hist simulations both calculated an aerosol TOA forcing of -
0.35 W m$^{-2}$ for the same period, slightly lower than recent estimates (e.g. -0.4 W m$^{-2}$ by
Markowicz et al., 2021). For the 2030-2050 period, the CMIP6 future ensemble simulated an
aerosol TOA forcing of -0.39±0.02 W m$^{-2}$, with SSP3-7.0 remaining unchanged compared to
the 1990-2010 mean (-0.35 W m$^{-2}$) and SSP1-2.6 and SSP2-4.5 becoming more negative (-
0.40 W m$^{-2}$). For the anthropogenic aerosols (Figure 7), the CMIP6 TOA forcing in 1990-
2010 is calculated to be -0.26 W m$^{-2}$, while in the 2030-2050 period the TOA anthropogenic
aerosol forcing became less negative (-0.25±0.03 W m$^{-2}$), being ~0.26 W m$^{-2}$ in SSP1-2.6 and
SSP2-4.5, and less negative in SSP3-7.0 (-0.21 W m$^{-2}$). Both ensembles estimated similar
TOA aerosol forcing compared to previous studies (e.g. Breider et al., 2017).
The different behaviour in the two ensembles is further investigated by looking at the forcing
calculated for the individual aerosol species of BC, OC, SO$_4^{2-}$ and NO$_3^-$ (Table 4 and Figure
8). The increase in cooling effect of aerosols calculated by the Eclipse ensemble is attributed
mainly to the decrease in BC as opposed to other aerosol species (Figure 8). More negative
forcing is calculated for the OC and NO$_3^-$, while the SO$_4^{2-}$ forcing is becoming less negative
due to large reductions in SO$_2$ emissions (Figure 1). The net aerosol forcing is therefore
slightly more negative. In the CMIP6 ensemble, the BC forcing does not change as much
compared to the Eclipse ensemble to counteract the change in impact from SO$_4^{2-}$, giving a
slightly more positive net aerosol forcing. The CMIP6 ensemble also simulates a larger
increase in the negative NO$_3^-$ forcing compared to the Eclipse ensemble (Shindell et al.,
2013). Overall, the changes in the different aerosol species leads to a higher negative aerosol
forcing by mid-century.
Overall, the Eclipse ensemble simulates slightly larger change in the aerosol forcings over the
2015-2050 period, based on the 1990-2010 mean, compared to the CMIP6 ensemble. These
changes are consistent with the changes in the aerosol burdens, where Eclipse simulations
calculated slightly larger changes in burdens compared to CMIP6 sinnulations. The Eclipse
ensemble simulation shows that the aerosol forcing (anthropogenic+natural) anomaly


becomes negative (-0.09±0.03 W m$^{-2}$) in 2050 compared to the 2015 anomaly (0.05±0.02 W
m$^{-2}$). The CMIP6 ensemble on the other hand shows that the 2050 anomaly becomes -
0.05±0.04 W m$^{-2}$.
3.4.    Climate change
*3.4.1. Surface air and sea surface temperatures*
The surface air temperature, precipitation, sea surface temperature and sea-ice extent are
calculated in the different simulations for the 1990-2050 period. As seen in Figure 9, the
Arctic surface air temperatures increase in all scenarios. Between 1990 and 2014, the surface
air temperatures over the Arctic increased by 5 °C decade$^{-1}$ (Eclipse_CplHist) to 10 °C
decade$^{-1}$ (CMIP6_Cpl_Hist), with a statistically significant ensemble mean trend of 7±2 °C
decade$^{-1}$. On the other hand, the observed surface air temperature during 1995-2014 shows a
smaller and statistically non-significant increase of 2 °C decade$^{-1}$. From 2015 onwards,
surface air temperatures continue to increase significantly by 5±1 °C decade$^{-1}$ in the Eclipse
simulations and by 4±1 °C decade$^{-1}$ in the CMIP6 simulations.
The Eclipse ensemble simulated an annual average surface temperature in the Arctic of -
7.44±0.94 °C in 1990 while the NINT-Cpl and CMIP6_Cpl_Hist simulated -8.32 °C and -
9.21 °C, respectively. The full ensemble simulated an annual average Arctic surface air
temperature of  -7.87±1.03 °C. The 2030-2050 mean surface air temperatures are projected to
increase by 2.1 °C and 2.4 °C compared to the 1990-2010 mean temperature (Figure 9)
according to the Eclipse CLE and MFR ensembles, respectively, while the CMIP6 simulation
calculated an increase of 1.9 °C (SSP1-2.6) to 2.2 °C (SSP3-7.0). These warmings are
smaller compared to the 4.5 - 5 °C warmer 2040 temperatures compared to the 1950-1980
average in the CMIP6 SSP1-2.6, SSP2-4.5 and SSP3-7.0 scenarios, reported by Davy and
Outen (2020). It should however be noted that due to the different baselines used in the
present study (1990-2010) and the 1950-1980 baseline used in Davy and Outen (2020), it is
not possible to directly compare these datasets. Figure 10 shows the spatial distributions of
the Arctic surface air temperature change between the 1990-2010 mean and the 2030-2050
mean for the individual Eclipse and CMIP6 future scenarios. All scenarios calculate a
warming in the surface air temperatures over the central Arctic, while there are differences
over the land areas. The Eclipse CLE and MFR ensembles show similar warming mainly
over the Arctic ocean as well as North America and North East Asia and cooling south of
Greenland. The latter is a well-known feature of observations and future projections, linked,
i.a., to the deep mixed layer in the area and declines in the Atlantic Meridional Circulation
(e.g. IPCC, 2014; Menary and Wood, 2018;  Keil et al., 2020). There are also differences
between the Eclipse and the CMIP6 ensembles as seen in Figure 10. All CMIP6 scenarios
show a warming over the central Arctic and a limited cooling over northern Scandinavia,
except for the SSP3-7.0 scenario that shows no cooling in the region. The SSP3-7.0-
lowNTCF scenario shows an additional cooling over Siberia. These warnings are comparable
with earlier studies, such as Samset et al. (2017) estimating a warming of 2.8 °C, attributed to
aerosols.





Following surface air temperatures, sea surface temperatures significantly increase ($p < 0.05$)
in all simulations (Figure 9). Between 1990-2014, the Eclipse simulations show a warming
trend of SSTs by $0.006 \pm 0.003$ °C yr$^{-1}$, while the CMIP6 simulations show a much larger
increase of 0.012 °C yr$^{-1}$. Both ensembles underestimated the observed SST trend of 0.017 °C
yr$^{-1}$. The Eclipse CLE and MFR scenarios predict a similar increase of 0.005 °C yr$^{-1}$, leading
to a slight increase of 0.25 °C in 2030-2050 mean surface air temperature compared to the
1990-2010 mean, while the CMIP6 simulations show an increase of $0.003 \pm 0.001$ °C yr$^{-1}$ ,
leading to an increase of 0.22 °C to 0.25 °C. Figure S2 shows the spatial distribution of the
sea surface temperature change between the 1990-2010 mean and the 2030-2050 mean for
the individual Eclipse and CMIP6 future scenarios. All simulations show a cooling of the sea
surface over the southern Greenland and north western Atlantic and a warming of the Pacific.
The Eclipse scenarios, in particular the MFR scenario, show a warming north of Europe,
while this warming is smaller in the CMIP6 simulations, except for the SSP3-7.0-lowNTCF
scenario that shows a comparable warming to the Eclipse CLE scenario.
*3.4.2. Sea-ice*
The Arctic sea-ice extent is found to decrease significantly in all simulations (Figure 9).
During the 1990-2014 period, the Eclipse ensemble simulated a decrease of  $34\,000 \pm 5\,800$
km$^2$ yr$^{-1}$, in agreement with the  observed decrease of 40 000 km$^2$ yr$^{-1}$, while
CMIP6_Cpl_Hist simulated a decrease of 70 000 km$^2$ yr$^{-1}$, largely overestimating the
observations. This overestimation has also been reported for some of the CMIP5 and CMIP6
models (Davy and Outten, 2020). After 2015, the Eclipse CLE ensemble projected a 37 000 $\pm$
12 000 km$^2$ yr$^{-1}$ decrease while the MFR simulated a slightly higher rate of decrease (-41 000
$\pm$ 5 000 km$^2$ yr$^{-1}$). The CMIP6 ensemble simulated a slightly smaller decrease rate (-27 000 $\pm$
8 000  km$^2$ yr$^{-1}$), with the largest decrease rate simulated by the SSP3-7.0 (-39 000 km$^2$ yr$^{-1}$).
The evolutions of March and September sea-ice extents are also analysed, representing the
Arctic annual maximum and minimum extents, respectively. The Eclipse ensemble projects a
decrease of 23 000 $\pm$ 11 000 km$^2$ yr$^{-1}$ in March sea-ice extent during the 2015-2050 period,
while the CMIP6 ensemble projects a decrease of 10 000 $\pm$ 6000 km$^2$ yr$^{-1}$ for the same
period, both statistically significant. In September, much larger decreases are projected by
both ensembles. The Eclipse ensemble simulates a decrease of 64 000 $\pm$ 10 000 km$^2$ yr$^{-1}$ in
the 2015-2050 period while the CMIP6 ensemble predicts a decrease of 50 000 $\pm$ 20 000 km$^2$
yr$^{-1}$.
The 2030-2050 annual mean sea-ice extent is projected to decrease by 1.5 and 1.7 million
km$^2$ compared to the 1990-2010 mean in the Eclipse CLE and MFR scenarios, respectively.
The CMIP6 simulations predict a lower decrease of sea-ice extent by 1.2 - 1.5 million km$^2$.
These results are comparable with the results from the CMIP6 models (Davy and Outten,
2020). In the 2030-2050 March mean the sea-ice extent is projected to decrease by 925 000
km$^2$ in the Eclipse ensemble, while the CMIP6 ensemble projects a decrease of 991 000 km$^2$.
A much larger decrease is projected for the 2030-2050 September mean, being 2.6 million
km$^2$ and 2.3 million km$^2$ in Eclipse and CMIP6 ensembles, respectively. As seen in Figure
11, the Eclipse ensemble predicts a decrease of September sea-ice fraction by up to 90% in a
band marking the maximum retreat of the sea ice line at the end of the summer. The CMIP6



SSP1-2.6 simulation shows a similar but higher decrease by up to 99%, while the SSP3-7.0
shows an increase of up to 95% over the Canadian Arctic and a decrease of up to 100% over
the Siberian Arctic, similar to the Eclipse ensemble. In March (Figure S1), all models agree
on a decrease in maximum sea-ice extent at the end of winter over the northern Pacific. In
addition, the Eclipse ensemble shows a decrease over the north Atlantic close to Greenland.
All simulations show a similar decrease in annual mean sea-ice extent (Figure S2) over the
central Arctic, with the CMIP6 ensemble showing also some increase in the sea-ice extent
over the Canadian Arctic, that is largest in SSP3-7.0. The retreat in sea-ice extent also led to
an increase of oceanic emissions of DMS and sea-salt (Figures S3-S7); however, the
increases (Figure S8) are not significant on a 5% significance level. The simulated increase,
in particular for the DMS emissions, is slightly larger in the Eclipse ensemble compared to
the CMIP6 ensemble, due to a larger decrease of sea-ice extent in the Eclipse ensemble. Also
note that GISS-E2.1 is using prescribed and fixed maps of DMS concentration in the ocean.
When ocean locations that are year-round under sea-ice at present get exposed, the DMS that
would exist in that sea water is not included in the simulations, likely underestimating the
increased flux of DMS into the atmosphere as the sea ice retreats.

4. Summary and Conclusions

The GISS-E2.1 earth system model has been used to simulate the recent past (1990-2014)
and future (2015-2050) aerosol burdens and their climate impacts over the Arctic. An
ensemble of seventeen simulations has been conducted, using historical and future
anthropogenic emissions and projections from CMIP6 and ECLIPSE V6b, the latter
supporting the ongoing Arctic Monitoring and Assessment Programme.

The evaluation of the recent past simulations shows underestimates of Arctic surface aerosol
levels by up to 50%, with the smallest biases calculated for the nudged AMIP-type Eclipse
simulations. An exception is $SO_4^{2-}$, where Eclipse_AMIP_NCEP had the highest bias, due to
the high cloud bias that leads to more in-cloud sulfate production from $SO_2$. The model skill
analyses indicate slightly better performance of the CMIP6 version of the GISS-E2.1 model
in simulating both the aerosol levels and climate parameters compared to the Eclipse version.
In addition, fully-coupled simulations had slightly smaller biases in aerosol levels compared
to atmosphere-only simulations (winds not nudged). Results from the various Eclipse
ensemble simulations showed that lowest biases in surface aerosols concentrations are
calculated for atmosphere only (prescribed sea-ice and sea-surface temperature) simulations
with nudged winds.

From 2015 onwards, all simulations show a statistically significant decrease in the Arctic BC,
OC and $SO_4^{2-}$ burdens, with the CMIP6 ensemble showing larger reductions in Arctic aerosol
burdens compared to the Eclipse ensemble. The Eclipse CLE and the CMIP6 SSP1-2.6 show
the largest reductions. Results indicated that the differences in burdens between the two
ensembles can be attributed to the different anthropogenic emissions datasets used. Results
from the various Eclipse simulations showed that the biogenic SOA contribution to the OC





burdens was higher in the nudged atmosphere only simulation, compared to the non-nudged
and coupled simulations.

The present-day (1990-2010 mean) CMIP6 and Eclipse simulations calculated an aerosol
TOA forcing of -0.35 W m$^{-2}$ and -0.32±0.01 W m$^{-2}$, respectively. For the same period, the
atmosphere only (AMIP) Eclipse simulations calculated a much larger aerosol TOA forcing
of -0.47 W m$^{-2}$. For the 2030-2050 period, both the Eclipse ensemble simulated an aerosol
TOA forcing of -0.39±0.01 W m$^{-2}$, of which -0.24±0.01 W m$^{-2}$ are attributed to the
anthropogenic aerosols (BC, OC, $SO_4^{2-}$ and $NO_3^-$). For the same period, the CMIP6 SSP3-7.0
simulated a similar TOA aerosol forcing (-0.35 W m$^{-2}$) compared to the 1990-2010 mean ,
while SSP1-2.6 and SSP2-4.5 scenarios simulated a more negative TOA forcing (-0.40 W m$^{-2}$
), of which the anthropogenic aerosols were responsible for -0.26 W m$^{-2}$). Overall, the
Eclipse ensemble simulated slightly larger changes in the aerosol forcings over the 2015-
2050 period, relative to the 1990-2010 mean, compared to the CMIP6 ensemble. The
differences between the two ensembles are further attributed to differences in the BC and
$SO_4^{2-}$ forcings. These results suggest that the different anthropogenic emission projections
between the two ensembles and within them lead to only small differences in how the aerosol
radiative forcing will evolve in the future over the Arctic.

The scenarios with the largest aerosol reductions, i.e. MFR in the Eclipse and SSP1-2.6 in the
CMIP6 ensemble projects a largest warming and sea-ice retreat. Overall, both Eclipse and
CMIP6 ensembles show a similar increasing trend of surface air temperatures over the Arctic
between 1990 and 2050, with the CMIP ensemble showing a slightly higher warming trend
(6±3 °C decade$^{-1}$) compared to the trend calculated by the Eclipse ensemble (5±1 °C decade$^{-1}$
). On the other hand, the Eclipse ensemble shows a slightly larger warming of 2030-2050
mean surface air temperatures of 2.1 to 2.5 °C over the Arctic compared to that from the
CMIP6 ensemble (1.9 °C to 2-2 °C). The Eclipse ensemble simulates a slightly larger
reduction in sea-ice extent in the Eclipse ensemble (-1.5 to -1.7 million km$^{-2}$ in CLE and
MFR, respectively) in 2030-2050 mean compared to the reduction in the CMIP6 scenario (-
1.3 to -1.6 million km$^{-2}$ in SSP1.2-6 and SSP3-7.0, respectively). However, the changes
simulated by the two ensembles are within one standard deviation of each other.

The overall results showed that the aerosol burdens will substantially decrease in the short- to
mid-term future, implying improvements in impacts on human health and ecosystems.,
Results also show that even the scenarios with largest emission reductions, i.e. Eclipse MFR
and CMIP6 SSP1-2.6, lead to similar impact on the future Arctic surface air temperatures
compared to scenarios with smaller emission reductions. On the other hand, scenarios with
very little mitigation such as the CMIP6 SSP3-7.0 leads to much larger sea-ice loss, implying
that even though impacts are small in temperatures, high mitigation of aerosols are still
necessary to limit sea-ice loss, exacerbating the dominant role played by well-mixed
greenhouse gases and underlining the importance of continued greenhouse gas reductions.
*Author contributions.* UI coordinated the study, conducted the model simulations, as well as
model evaluation and analyses of the simulations, and wrote the manuscript. KT and GF
supported the model simulations and processing of the Eclipse V6b emissions for the GISS-



E2.1 model. JPF contributed to the plotting of the spatial distributions by further developing
the autoimage R package (French, 2017). RM prepared and provided the AOD
measurements, as well as the surface air temperature, sea surface temperature and sea-ice
data. TM prepared the cloud observation data. CHW prepared the Arctic surface aerosol
measurement data. KvS coordinated the experimental setup for the Eclipse simulations in the
framework of the ongoing AMAP assessment. ZG prepared and provided the Eclipse V6b
anthropogenic emissions. HS and DCT prepared the Villum Research Station aerosol data. JB
and PL contributed to analyses of aerosols and climate parameters, respectively, and
manuscript writing. All authors contributed to the analyses and interpretation of the results, as
well as contributing to the writing of the manuscript.
*Competing Interests.* The authors declare that they have no conflict of interest.
*Special issue statement.* This article is part of the special issue "Arctic climate, air quality,
and health impacts from short-lived climate forcers (SLCFs): contributions from the AMAP
Expert Group".
*Acknowledgements.* This paper was developed as part of the Arctic Monitoring Assessment
Programme (AMAP), AMAP 2021Assessment: Arctic climate, air quality, and health
impacts from short-lived climate forcers (SLCFs). HadISST data were obtained from
https://www.metoffice.gov.uk/hadobs/hadisst/ and are © British Crown Copyright, Met
Office, provided under a Non-Commercial Government Licence
http://www.nationalarchives.gov.uk/doc/non-commercial-government-licence/version/2/.
UDel_AirT_Precip data provided by the NOAA/OAR/ESRL PSL, Boulder, Colorado, USA,
from their Web site at https://psl.noaa.gov/. Alert sulfate data are from Sangeeta and EC &
OC data from Lin Huang, respectively, as part of Canadian Aerosol Baseline Measurement
(CABM) program at ECCC and would like to thank operators & technicians for collection of
filters, calibration and analysis and Canadian Forces Services Alert for the operation of the
military base. These datasets are also available on Global Atmospheric Watch program,
World Data Center for aerosols, EBAS database (http://ebas.nilu.no/default.aspx).Aside from
Alert, Canada's surface air quality data are from the National Atmospheric Pollutant
Surveillance network (NAPS: https://open.canada.ca/data/en/dataset/1b36a356-defd-4813-
acea-47bc3abd859b).
Fairbanks aerosol measurements are from William Simpson and KC Nattinger. Aside from
Fairbanks, Alaskan measurements are from the IMPROVE network. IMPROVE is a
collaborative association of state, tribal, and federal agencies, and international partners. The
US Environmental Protection Agency is the primary funding source, with contracting and
research support from the National Park Service. The Air Quality Group at the University of
California, Davis is the central analytical laboratory, with ion analysis provided by Research
Triangle Institute, and carbon analysis provided by Desert Research Institute.
European measurements are from the EMEP network, and obtained from the EBAS database
(http://ebas.nilu.no). Other European data include the Gruvebadet measurements, for which
we acknowledge Mauro Mazzola (mauro.mazzola@cnr.it), Stefania Gilardoni
(stefania.gilardoni@cnr.it), and Angelo Lupi (angelo.lupi@cnr.it) from the Institute of Polar
Sciences fo Gruvabadet eBC measurements; and Rita Traversi (rita.traversi@unifi.it), Mirko



Severi (mirko.severi@unifi.it), and  Silvia Becagli (silvia.becagli@unifi.it) from University
of Florence http://www.isac.cnr.it/~radiclim/CCTower/ ?Data:Aerosol; the Zeppelin datasets,
for which we acknowledge Vito Vitale and Angelo Lupi (also available on
http://ebas.nilu.no); and the Villum Station datasets (www.villumresearchstation.dk) from
Henrik Skov (hsk@envs.au.dk; also available in http://ebas.nilo.no). The AERONET AOD
measurements were obtained from NASA's Goddard Space Flight Center
(https://aeronet.gsfc.nasa.gov/new_web/index.html). The authors acknowledge Dr. L.
Sogacheva and AEROSAT team for satellite based merged AOD data.

*Financial support.* This research has been supported by the Aarhus University
Interdisciplinary Centre for Climate Change (iClimate) OH fund (no. 2020-0162731), the
FREYA project, funded by the Nordic Council of Ministers (grant agreement no. MST-227-
00036 and MFVM-2019-13476), and the EVAM-SLCF funded by the Danish Environmental
Agency (grant agreement no. MST-112-00298). KT and GF thank the NASA Modeling,
Analysis and Prediction program (MAP) for support. ZK was financially supported by the
EU-funded Action on Black Carbon in the Arctic (EUA-BCA) under the EU Partnership
Instrument. JPF was partially supported by NSF award 1915277.

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

October 26[th])



**Tables**

Table 1. GISS-E2.1 simulations carried out in the Eclipse and CMIP6 ensembles.

| Simulations | Description | No. Ensemble | Period |
|---|---|---|---|
| NINT_Cpl | No tracers- Coupled | 1 | 1850-2014 |
| Eclipse_AMIP | AMIP OMA | 1 | 1995-2014 |
| Eclipse_AMIP_NCEP | AMIP OMA – winds nudged to NCEP | 1 | 1995-2014 |
| Eclipse_CplHist | OMA – Coupled | 3 | 1990-2014 |
| Eclipse_Cpl_CLE | OMA – Coupled | 3 | 2015-2050 |
| Eclipse_Cpl_MFR | OMA – Coupled | 3 | 2020-2050 |
| CMIP6_Cpl_Hist | OMA – Coupled | 1 | 1850-2014 |
| CMIP6_Cpl_SSP1-2.6 | OMA – Coupled | 1 | 2015-2050 |
| CMIP6_Cpl_SSP2-4.5 | OMA – Coupled | 1 | 2015-2050 |
| CMIP6_Cpl_SSP3-7.0 | OMA – Coupled | 1 | 2015-2050 |
| CMIP6_Cpl_SSP3-7.0-lowNTCF | OMA – Coupled | 1 | 2015-2050 |

Table 2. Annual mean Normalised Mean Bias (*NMB:%*) and correlation coefficients (*r*) for the recent past simulations in the GISS-E2.1 model ensemble during 1995-2014 for BC, OC, $SO_4^{2-}$ and 2008/2009-2014 for AOD550 from AERONET and satellites.

| | BC | | OC | | $SO_4^{2-}$ | | AOD_aero | | AOD_sat | |
|---|---|---|---|---|---|---|---|---|---|---|
| Model | *NMB* | *r* | *NMB* | *r* | *NMB* | *r* | *NMB* | *r* | *NMB* | *r* |
| AMAP_OnlyAtm. | -67.32 | 0.27 | -35.46 | 0.54 | -49.83 | 0.65 | -33.28 | -0.07 | -0.48 | 0.00 |
| AMAP_OnlyAtm_NCEP | -57.00 | 0.26 | -7.80 | 0.56 | -52.70 | 0.74 | -41.99 | 0.02 | -0.55 | 0.13 |
| AMAP_CplHist1 | -62.82 | 0.21 | -22.85 | 0.51 | -50.13 | 0.70 | -47.42 | 0.03 | -0.59 | -0.00 |
| AMAP_CplHist2 | -63.49 | 0.29 | -17.99 | 0.63 | -48.44 | 0.71 | -41.89 | 0.01 | -0.55 | 0.1 |
| AMAP_CplHist3 | -62.70 | 0.27 | -16.36 | 0.60 | -49.60 | 0.70 | -40.53 | 0.07 | -0.53 | 0.11 |
| CMIP6_Cpl_Hist | -49.90 | 0.26 | 13.14 | 0.69 | -39.81 | 0.70 | -39.86 | 0.05 | -0.53 | 0.11 |



Table 3. Annual mean biases and correlation coefficients (*r*) for the recent past simulations in the GISS-E2.1 model ensemble in 1995-2014 for surface air temperature (Tsurf) and sea surface temperature (SST) in units of °C, and precipitation (Precip), sea-ice fraction (Sea-ice), total cloud fraction (CldFrc), liquid water path (LWP), and ice water path (IWP) in units of %.

| Simulations | IWP | | LWP | | Cld Frac | | Sea-ice | | SST | | Precip | | Tsurf | |
|---|---|---|---|---|---|---|---|---|---|---|---|---|---|---|
| | *r* | *NMB* (%) | *r* | *NMB* (%) | *r* | *NMB* (%) | *r* | *NMB* (%) | *r* | *MB* (°C) | *r* | *NMB* (%) | *r* | *MB* (°C) |
| NINT | 0.53 | -56.06 | -0.89 | 70.55 | -0.67 | 20.95 | 1.00 | 12.14 | 0.99 | -1.96 | 0.88 | -52.68 | 1.00 | -0.22 |
| AMAP_OnlyAtm. | -0.18 | -58.53 | -0.96 | 57.52 | -0.81 | 23.78 | 1.00 | -2.56 | 0.99 | -1.50 | 0.89 | -50.33 | 1.00 | 0.55 |
| AMAP_OnlyAtm_NCEP | -0.64 | -70.32 | -0.91 | 14.19 | -0.79 | 24.83 | 1.00 | -2.56 | 0.99 | -1.50 | 0.90 | -53.19 | 1.00 | 0.35 |
| AMAP_CplHist1 | 0.56 | -55.38 | -0.90 | 72.60 | -0.66 | 21.63 | 1.00 | 11.04 | 0.99 | -1.93 | 0.87 | -52.63 | 1.00 | -0.04 |
| AMAP_CplHist2 | 0.44 | -56.53 | -0.93 | 68.63 | -0.65 | 21.48 | 0.99 | 11.13 | 0.99 | -1.92 | 0.84 | -53.96 | 1.00 | -0.09 |
| AMAP_CplHist3 | 0.45 | -55.32 | -0.91 | 71.75 | -0.66 | 21.79 | 1.00 | 11.88 | 0.99 | -1.94 | 0.86 | -52.59 | 1.00 | -0.14 |
| CMIP6_Cpl_Hist | 0.40 | -56.28 | -0.91 | 69.18 | -0.65 | 21.49 | 0.99 | 12.56 | 0.98 | -1.96 | 0.85 | -53.96 | 1.00 | -0.17 |




Table 4. TOA aerosol radiative forcings for 1990-2010 and 2030-2050 periods as calculated by the GISS-E2.1.

| Simulations | SSA | | Dust | | NO3 | | SO4 | | OC | | BC | | Aerosols Anth. | | Aerosols Total | |
|---|---|---|---|---|---|---|---|---|---|---|---|---|---|---|---|---|
| | 2030-2050 | 1990-2010 | 2030-2050 | 1990-2010 | 2030-2050 | 1990-2010 | 2030-2050 | 1990-2010 | 2030-2050 | 1990-2010 | 2030-2050 | 1990-2010 | 2030-2050 | 1990-2010 | 2030-2050 | 1990-2010 |
| NINT_Cpl | | -0.23 | | 0.06 | | -0.01 | | -0.33 | | -0.05 | | 0.2 | | -0.19 | | -0.35 |
| Eclipse_AMIP | | -0.27 | | 0.09 | | -0.02 | | -0.39 | | -0.06 | | 0.2 | | -0.27 | | -0.46 |
| Eclipse_AMIP_NCEP | | -0.23 | | 0.08 | | -0.04 | | -0.39 | | -0.08 | | 0.19 | | -0.32 | | -0.47 |
| Eclipse_CplHist_3xEns | | -0.22 | | 0.12 | | -0.03 | | -0.38 | | -0.05 | | 0.23 | | -0.22 | | -0.32 |
| Eclipse_CplCLE_3xEns | -0.24 | | 0.09 | | -0.07 | | -0.27 | | -0.07 | | 0.17 | | -0.24 | | -0.39 | |
| Eclipse_CplMFR_3xEns | -0.25 | | 0.09 | | -0.04 | | -0.22 | | -0.07 | | 0.09 | | -0.23 | | -0.39 | |
| CMIP6_Cpl_Hist | | -0.21 | | 0.12 | | -0.04 | | -0.4 | | -0.06 | | 0.23 | | -0.26 | | -0.35 |
| CMIP6_Cpl_SSP126 | -0.24 | | 0.09 | | -0.1 | | -0.22 | | -0.07 | | 0.13 | | -0.26 | | -0.4 | |
| CMIP6_Cpl_SSP245 | -0.23 | | 0.09 | | -0.09 | | -0.29 | | -0.08 | | 0.19 | | -0.27 | | -0.41 | |
| CMIP6_Cpl_SSP370 | -0.23 | | 0.09 | | -0.06 | | -0.34 | | -0.09 | | 0.28 | | -0.21 | | -0.35 | |
| CMIP6_Cpl_SSP370-lowNTCF | -0.23 | | 0.09 | | -0.09 | | -0.28 | | -0.07 | | 0.2 | | -0.24 | | -0.38 | |





**Figures**

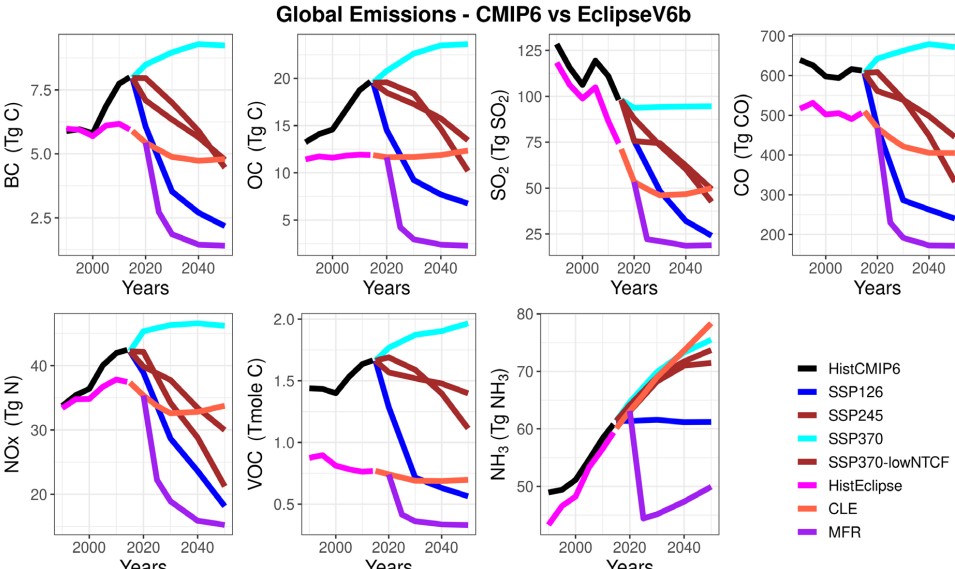

Figure 1. Global recent past and future CMIP6 and Eclipse V6b anthropogenic emissions for different pollutants and scenarios.





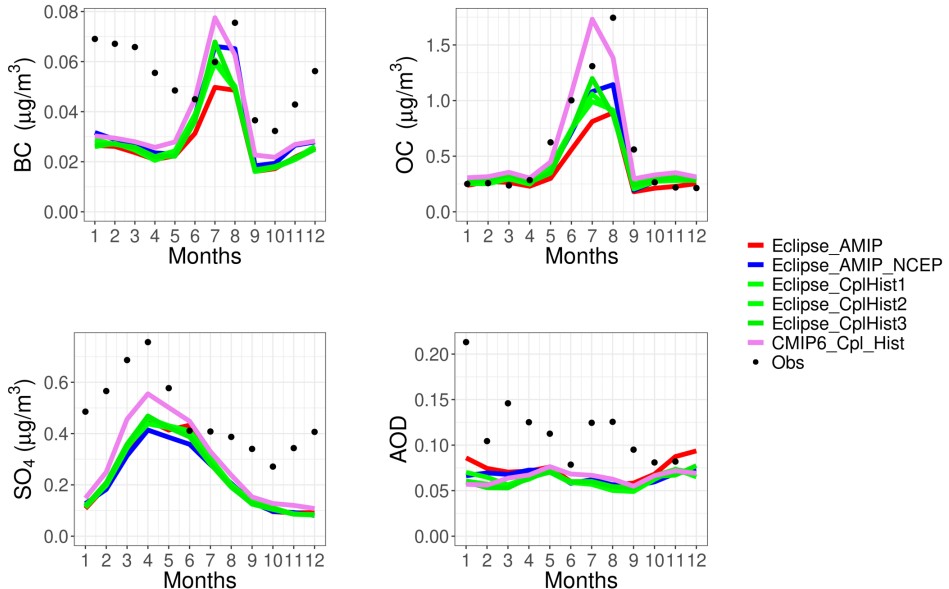

Figure 2. Observed and simulated Arctic climatological (1995-2014) monthly BC, OC, $SO_4^{2-}$, and AERONET AOD at 550nm (2008/09-14).

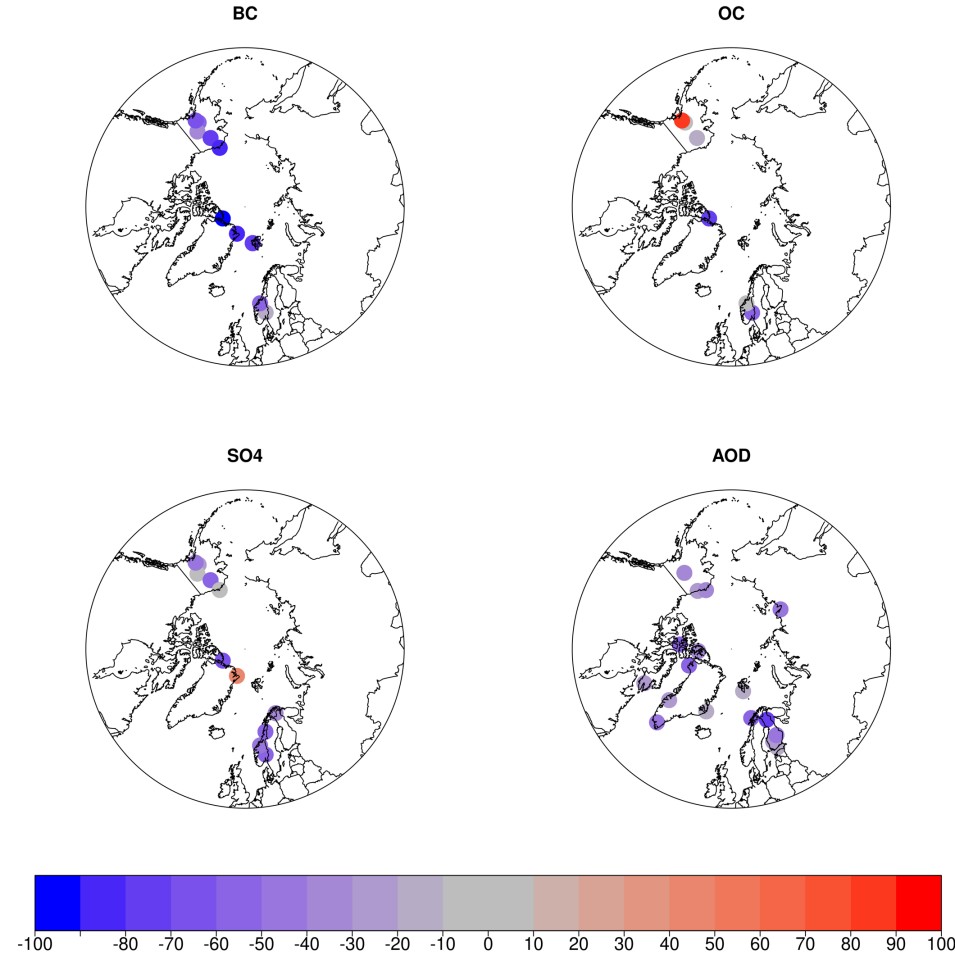

Figure 3. Spatial distribution of normalized mean bias (*NMB*, in %) for climatological mean
(1995-2014) BC, OC, SO₄²⁻ and AOD at monitoring stations, calculated as the mean of all
recent past simulations.





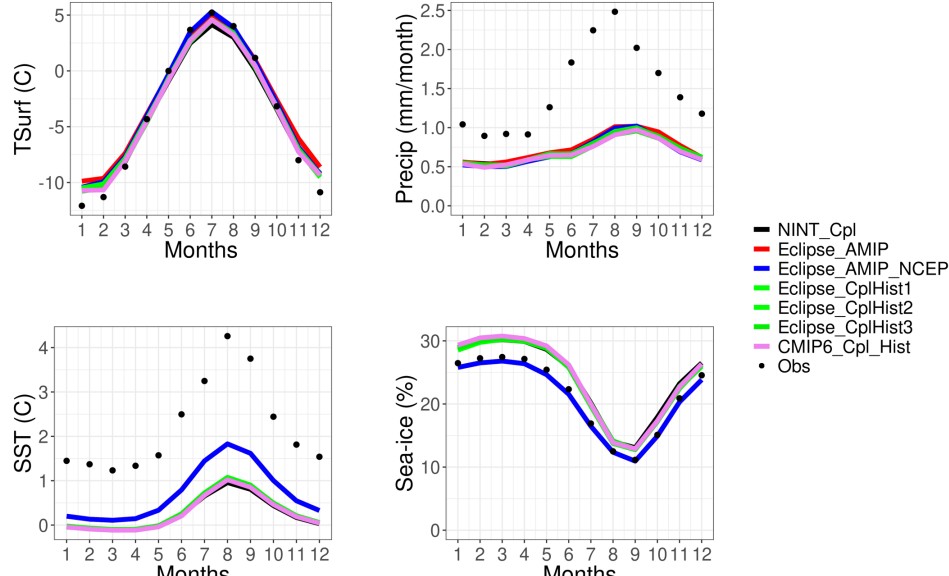

Figure 4. Observed and simulated Arctic climatological (1995-2014) surface air temperature, precipitation, sea surface temperature, and sea-ice. Obs denote UDel dataset for surface air temperature and precipitation, and HADISST for sea surface temperature and sea-ice extent. Note that the two AMIP runs (blue and red lines) for the SST and sea-ice are on top of each other as they use that data to run, as input.



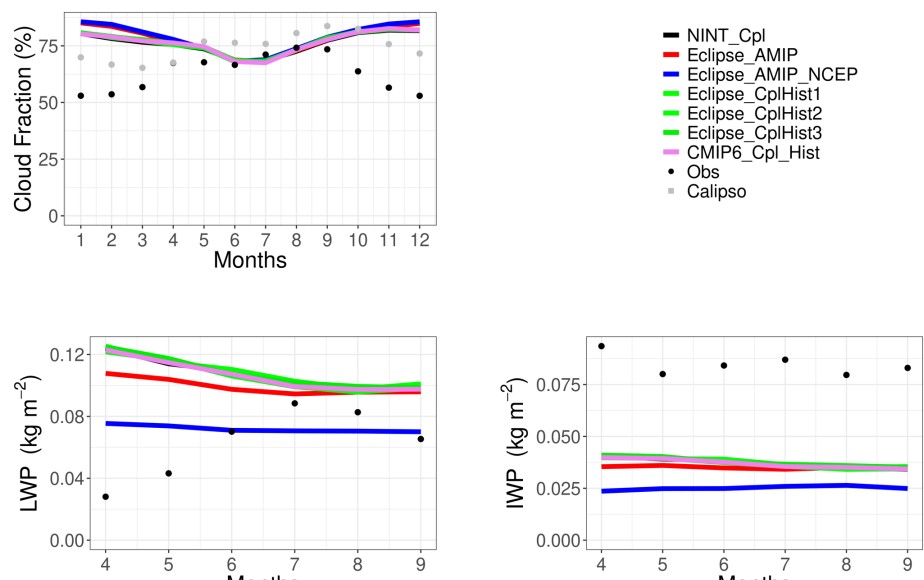

Figure 5. Observed and simulated Arctic climatological total cloud fraction (1995-2014 mean), liquid water path (2007-2014 mean), and ice water path (2007-2014 mean). Obs denote Clara-A2 for the cloud fractions and CloudSat for the LWP and IWP.





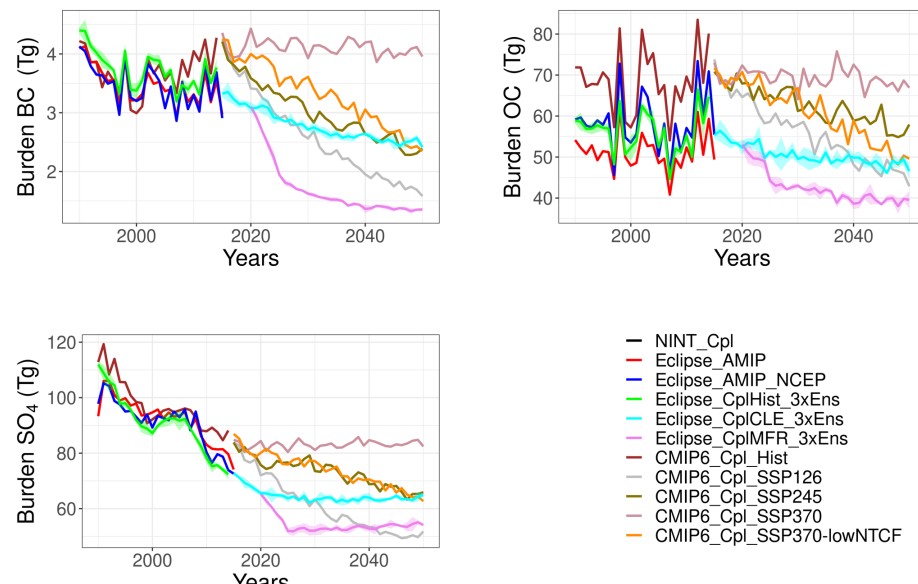

Figure 6. Arctic BC, OC and SO$_4^{2-}$ burdens in 1990-2050 as calculated by the GISS-E2.1 ensemble.





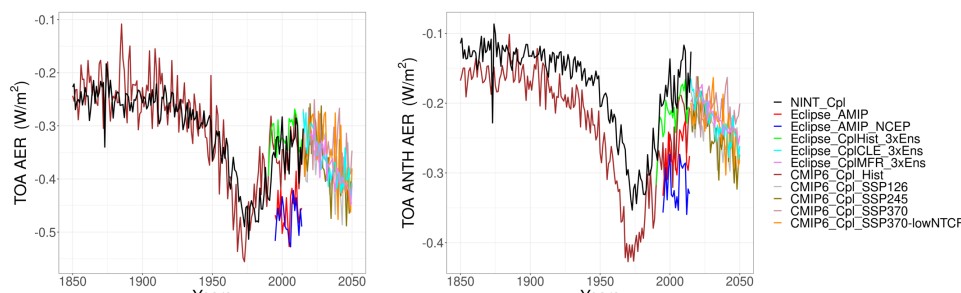

Figure 7. Arctic TOA aerosol radiative forcing from anthropogenic and natural aerosols $(BC+OC+SO_4^{2-}+NO_3^-+Dust+SSA)$, and only anthropogenic aerosols $(BC+OC+SO_4^{2-}+NO_3^-)$ in 1850-2050 as calculated by the full GISS-E2.1 ensemble.





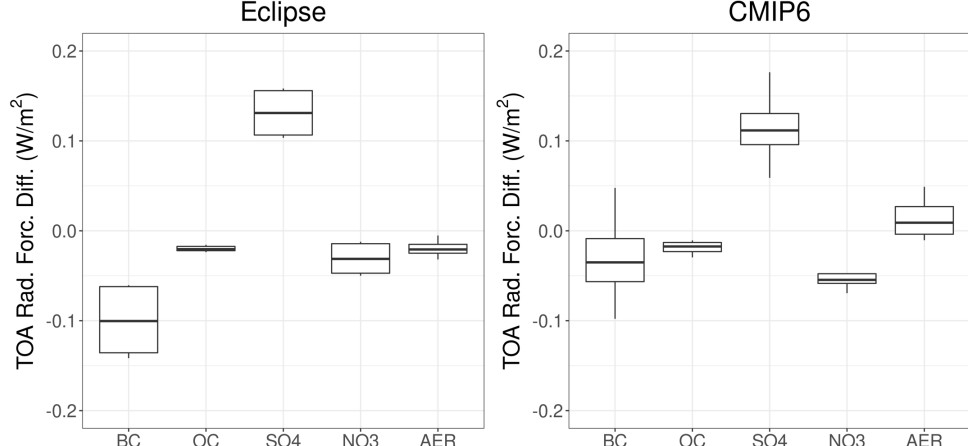

Figure 8. Box-Whisker plot showing the differences between 1990-2010 mean and 2030-2050 mean TOA radiative forcing for the anthropogenic aerosol components (BC, OC, $SO_4^{2-}$ and $NO_3^-$) and their sum (AER) in the Eclipse (left panel) and the CMIP6 (right panel) ensembles. The boxes show the median, the 25th and 75th percentiles. The upper whisker is located at the *smaller* of the maximum value and $Q\_3 + 1.5$ IQR, whereas the lower whisker is located at the *larger* of the smallest x value and $Q\_1 – 1.5$ IQR, where IQR (interquartile range) is the box height (75th percentile - 25th percentile).



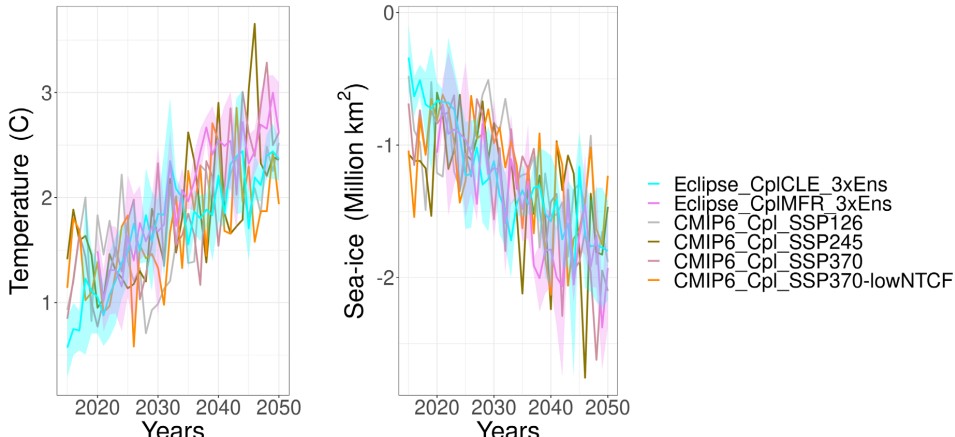

Figure 9. Arctic annual mean surface air temperature and sea-ice extent anomalies in 2015-2050 based on the 1990-2010 mean as calculated by the GISS-E2.1 ensemble.


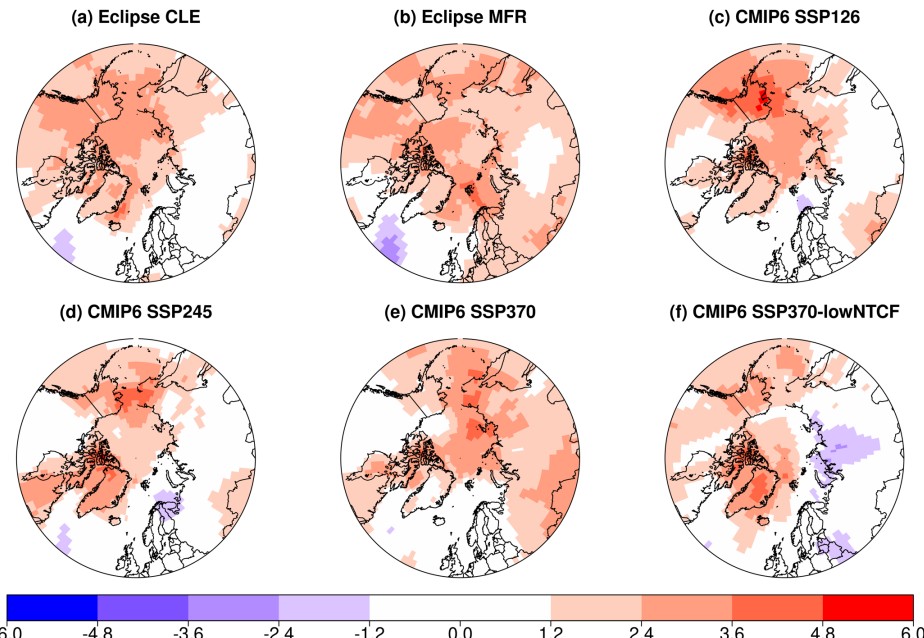

Figure 10. Spatial distribution of the annual mean Arctic surface air temperature (°C) changes between the 1990-2010 mean and the 2030-2050 mean as calculated by the GISS-E2.1 ensemble.





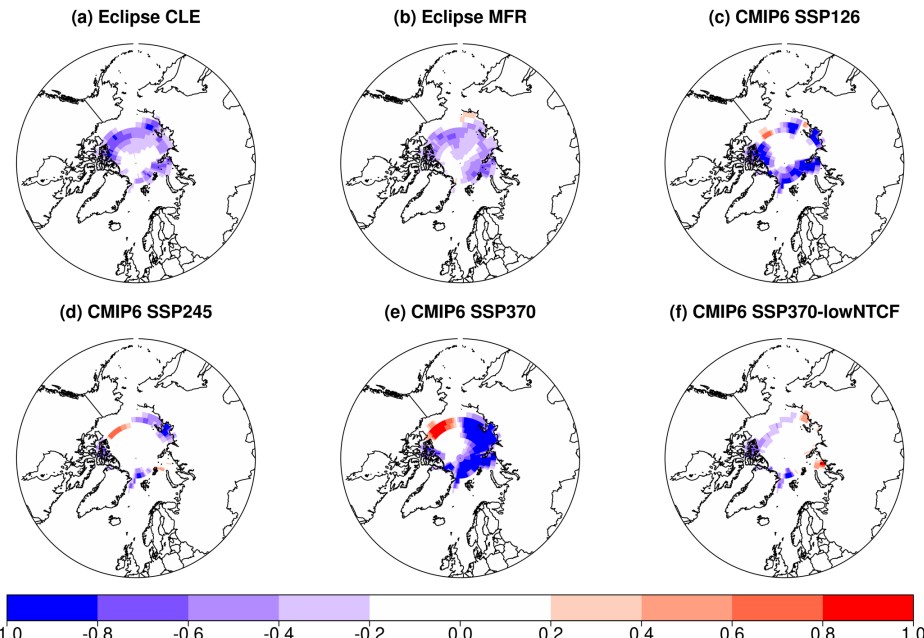

Figure 11. Spatial distribution of the September Arctic sea-ice fraction change between the 1990-2010 mean and the 2030-2050 mean as calculated by the GISS-E2.1 ensemble.