# Peer review of "Present and future aerosol impacts on Arctic climate change in the GISS-E2.1 Earth system"

_Atmospheric Chemistry and Physics, 2020_

## Referee Comment (RC1)

**Review for "Present and future aerosol impacts on Arctic climate change in the GISS-E2.1 Earth system model"**

By Ulas Im et al.

**Summary**

The manuscript reports on an extensive set of simulations with the GISS-E2.1 Earth system model spanning both near past and future, which investigate the impact of changing anthropogenic aerosol emissions on Arctic climate. The study is interesting and in principle suitable for ACP, but in places I find the text quite hard to follow to the point that I am not sure whether the results support the conclusions. However, this is mainly due to the presentation of the results (both in text and figures), not the results themselves and I would be happy to review a revised version of the manuscript again.

**General comments**

The terms used in the radiative forcing discussion are somewhat outdated. In AR5, the IPCC recommends moving from direct and indirect effects to using radiative forcing due to aerosol-radiation interactions ($RF_{ARI}$) and due to aerosol-cloud interactions ($RF_{ACI}$), together with their resulting rapid adjustments and the final effective radiative forcing (see chapter 7 of AR5). In order to keep up with this development, I strongly recommend rewriting the text with respect to this.

Many parts of the results section contain long listings of changes of different quantities in the different scenarios for several time periods and are quite hard to follow. I'm wondering whether it would be more beneficial to organise these results in tables and rather concentrate on systematic or principle differences between the different simulations. For instance, if there is a systematic decrease in sulfate emissions in the ssp simulations, how does this translate into Arctic sulfate burdens, radiative forcings and temperatures and how are the results of the Eclipse simulations different from that?

In the Discussion section I am missing a discussion on how the biases that have been found in the model evaluation section may affect the modelled climate impacts in the future and if and how much that adds to the uncertainty of the results.

**Specific comments**

**Abstract**

- lines 30—32: add "In the simulations" to "Surface aerosol levels ... have been significantly underestimated"
- line 32: "The nudged simulations" have not been defined at this point. I recommend changing this to "...when winds were nudged to reanalysis data"
- line 34: A change from "fully coupled simulations" to "simulations where atmosphere and ocean where coupled' or something similar might be better at this point.

- lines 37—48: None of the simulation names have been introduced at this point (naturally) and it might therefore be hard for the potential reader to grasp the general message of the abstract. I therefore recommend to re-write this paragraph. In my opinion a "maximum vs minimum effect"-type of discussion would be easier to digest at this point.
- line 46: remove "both"
- line 46—47: Change "In 2050" to "By 2050"?
- line 52: "while scenarios no or little…" – add "with"
- line 53: "lead" --> "leads"

**Introduction**

- line 71: "This contribution … puts" or "These contributions … put" ?
- lines 80 --- 85: "BC" and "$SO_4^{2-}$" have already been defined.
- line 90: I'm not sure myself: Is BC depositing on snow and ice or is BC being deposited, e.g. can you use the active form here?
- line 93: While you talk about the lifetime and vertical extent effects here, if I understand the model description correctly, these effects are not included in your simulation, or are they?
- lines 111—112: Is that global emissions?

**Materials and Methods**

- line 169—174: Can you elaborate on how that works? If Everything except dust and sea salt is externally mixed, does that mean that the model assumes separate sulphate, nitrate, BC and OC particles? How do you then treat the sulphate and nitrate coating of the dust particles?
- Even though SOA production in the model is described in Tsigaridis and Kanakidou, maybe you could describe it briefly here as well. In particular, what are the assumptions of how SOA formation affects OC concentrations. This is important, as you attribute higher OC concentrations to higher SOA formation, but it is not clear, how that is modelled. Do you have separate SOA tracers or does VOC oxidation lead directly to OC production in the atmosphere? In the former case, how do you convert SOA into OC. Am I right in assuming that OC from the emission inventories is emitted as particulate matter?
- line 178—180: How does that work? If the model treats the first indirect (i.e. aerosol concentrations affecting CDNC and (I guess) cloud droplet size), how do you stop the model from changing LWP and precipitation rates?
- line 186: I guess this is also just the first indirect effect?
- Section 2.2.3: Do I understand this correctly: Eclipse emissions have been complemented in some sectors by using CEDS emissions, while CEDS emissions are entirely "original", or did you also have to complement CEDS emissions in some sectors?
- Lines 284 – 288: You have been quite thorough in explaining the differences between the ECLIPSE scenarios, but the differences between the different CEDS scenarios is quite compact. What, for instance, does "lowNTCF" mean?
- Section 2.2.3: How do the emissions and concentrations of Greenhouse gases evolve in the simulations? Are they kept fixed to capture the aerosol effect, or do they change? In the latter case, please elaborate on how you separate the aerosol effects from the Greenhouse gas effects.

- Section 2.2.3: If emissions are provided at 0.5x0.5° resolution, but the model operates at 2x2.5° resolution, I'm guessing you re-grid the emissions somehow?
- lines 317—326: As a side note, it has become more and more common to co-locate modelling data and observations in time to reduce the effects of observational "data sparseness" mentioned here. I understand that this is probably out of scope of this study, but worth considering in the future.

**Results**

- Figure 3: It would be quite beneficial to add the station names to the figure. Especially because some of the stations are discussed in the text.
- lines 395-398: Could these high bias outliers be a problem with the representativeness of the observations (e.g. too few data points, or quickly changing orography)? Trapper Creek, for instance, is right next to another, blue, point.
- lines 433—436: Later in the article (line 774) you state that a higher cloud fraction may lead to higher in-cloud SO4 production – please add this statement also here.
- Tables 3 and 4: Please consider breaking up these tables into two parts and displaying them in portrait mode. At least in electronic form it would make the manuscript easier to read.
- Why do the AMIP runs have such a high bias in SST, if SST is prescribed?
- lines 461—462: Is that due to model resolution? After all, SIC is prescribed, right?
- lines 470—471: Do you mean the climatology of the cloud fraction for the entire year here?
- Figure 5a: I think here it would be worth mentioning that the seasonal trends in observed and modelled cloud fraction trends are reversed. Looking at panels b and c, it almost looks like the model produces to few water or mixed-phase clouds during the winter months, did I get this right?
- lines 474 – 478: This sentence is very hard to grasp: Less overestimation due to an underestimation? Do you mean to say that you trust AVHRR CLARA-A2 less than CALIPSO, because CALIPSO does a better job at separating bright surfaces from clouds? Also, you could add in line 466 that there you compare to AVHRR data.
- Figure 6 and Section 6.2: I take it that by Arctic burden you mean the integral over all grid boxes between 60 and 90° north and over all vertical levels, but then using monthly averages? Why do you use the unit kTon in the text, but Tg in the figures?
- line 533: What do you mean by "better resolved"?
- lines 541-542: If you term it "reduction", I guess the number should be positive...
- line 549 and following: How has statistical significance been tested?
- line 554—555: See my comment in the Materials and Methods section. If OC is a separate tracer, you should explain somewhere, how a larger SOA production leads to larger OC concentrations. If it is what I think (i.e. you talk the sum of OC and SOA species), I suggest calling it something else. Maybe organic aerosol (OA) or organic matter (OM) would be suitable?
- Figure S1: This links directly to the comment above. Without any explanation, it is not really understandable what you are showing here.
- line 580: ...because CLE levels off earlier (no further legislation after this point?). The calculated trend cannot really be 2015—2050.
- Figure 7: What are you actually plotting here? From the explanation in the text (double call to the radiation code with and without aerosols) it sounds like you are

showing the radiative effect due to aerosol-radiation interaction ($RE_{ARI}$) (see Chapter 7.3.4.1 of the IPCC AR5), formerly termed the "direct radiative effect". A radiative forcing due to aerosol-radiation interaction ($RF_{ARI}$) would be the change in $RE_{ARI}$ relative to some reference point, e.g. preindustrial levels. Please elaborate.

- line 595 – 602: Why do you only talk about Eclipse here?
- 595 – 597: Why is that? This is quite a substantial difference – can this be explained by differences in aerosol burdens alone?
- line 601—602: What is the meaning of the third value here?
- Figure 7: Why do the AMIP runs differ so much from the other simulations (2000—2015)? Also, there is visible difference between the black and brown lines (NINT_Cpl and CMIP6_Cpl_Hist?) in the anthropogenic aerosol radiative forcings, byt the same difference is not visible for the total aerosol radiative forcing – what is compensating for the difference here?
- If SOA can contribute to OC and if SOA can originate from both natural and anthropogenic sources, how can you separate the anthropogenic contribution of OC to the radiative forcing?
- Figure 8: how are the speciated forcings calculated?
- lines 640—651: This appears to be exactly the same text as lines 604—615.
- line 650: What is higher to what here?
- line 656: "sinnulations" --> "simulations"
- line 657: You use the term "anomaly" the first time here – how is this calculated and what do you mean by "aerosol forcing anomaly"?
- Figure 9: In the figure you show only the surface temperatures between 2020 and 2050, but you talk a lot about temperature trends in earlier times – is there a reason for this? Also, it would be much easier to follow the discussion, if the observed trends would be added to the figure.
- Lines 665 – 673 I can't really believe the numbers you give in this paragraph. A 10°C/decade increase in surface air temperature is huge, even for the Arctic. As a reference, in the Figure 9 you show the surface temperatures between 2020 and 2050, which change by about 1-2°C in three decades. Please check your calculations or provide a figure, if the numbers are correct.
- 698: "warnings" --> "warmings"?
- Figure 10: How statistically robust are these spatial distributions? Looking only at the SSP results (panels c, d and e), it looks like the changes are not very systematic in many regions, which makes me wonder how noisy the results are.
- line 702: Figure 9 does not show SST.
- line 712: Do you mean "Greenland sea"?
- line 736: Here and in some other places where you compare the means of two time periods, you could consider replacing "... is projected to decrease by ... compared to ..." with "... is projected to be ... lower than ..."
- Figure 11: Even though I the discussion is generally about the entire Arctic region, in this figure I'm wondering if it would be better to "zoom in" to where the changes are actually happening.
- line 748: "Figure S1" --> "Figure S3"
- line 751: "Figure S2" shows SST
- line 754: "Figures S3—S7" --> "Figures S4—S7"

**Summary and Conclusions**

- line 773: Like in the abstract, I would try to avoid using the names of the individual simulations in the conclusions.
- line 808: add "future"
- lines 815 – 818. There appears to be one "Eclipse" too much.
- line 826 – 829: Could one interpret this as the melting of sea ice acting buffering the changes in surface air temperatures?

---

## Author Comment (AC1)

**Response to Reviewers**

We would like to thank both the reviewers for the valuable comments and suggestions. We have tried to implement all the suggestions proposed by the reviewers and hope that the new version is structured in a better way to ease the readers and is suitable for publication in ACP.

**Reviewer 1**

Summary

The manuscript reports on an extensive set of simulations with the GISS-E2.1 Earth system model spanning both near past and future, which investigate the impact of changing anthropogenic aerosol emissions on Arctic climate. The study is interesting and in principle suitable for ACP, but in places I find the text quite hard to follow to the point that I am not sure whether the results support the conclusions. However, this is mainly due to the presentation of the results (both in text and figures), not the results themselves and I would be happy to review a revised version of the manuscript again.

*Response: We thank the reviewer for the general positive response to the manuscript. We have restructured the manuscript, as well as the presentation of the results and discussion.*

General comments

The terms used in the radiative forcing discussion are somewhat outdated. In AR5, the IPCC recommends moving from direct and indirect effects to using radiative forcing due to aerosol-radiation interactions (RFARI) and due to aerosol-cloud interactions (RFACI), together with their resulting rapid adjustments and the final effective radiative forcing (see chapter 7 of AR5). In order to keep up with this development, I strongly recommend rewriting the text with respect to this.

*Response: The new version now is using the $RF_{ARI}$ (Section 3.2)*

Many parts of the results section contain long listings of changes of different quantities in the different scenarios for several time periods and are quite hard to follow. I'm wondering whether it would be more beneficial to organise these results in tables and rather concentrate on systematic or principle differences between the different simulations. For instance, if there is a systematic decrease in sulfate emissions in the ssp simulations, how does this translate into Arctic sulfate burdens, radiative forcings and temperatures and how are the results of the Eclipse simulations different from that?

*Response: We thank the reviewer for the suggestion. We now have more tables in the manuscript and tried to focus more on the differences between different scenarios (Section 3.2). However, while we can directly connect the forcings to burdens of individual aerosol components, it is not possible to further extent this to climate impacts per species as this requires explicit sensitivity simulations.*

In the Discussion section I am missing a discussion on how the biases that have been found in the model evaluation section may affect the modelled climate impacts in the future and if and how much that adds to the uncertainty of the results.

*Response: We have now added the following sentence in section 3.1.2 (Lines 518-524): "Results show that both absorbing (BC) and scattering aerosols (OC and $SO_4^{2-}$) are underestimated by the GISS-E2.1 model, implying that these biases can partly cancel out their impacts on radiative forcing due to aerosol-radiation interactions. This, together with the very low biases in surface temperatures suggests that aerosols over the Arctic do not affect the Arctic climate and that the changes in Arctic climate are mainly driven by changes due to greenhouse gas concentrations.".*

Specific comments

Abstract
• lines 30—32: add "In the simulations" to "Surface aerosol levels ... have been significantly underestimated"
*Response: Sentence is changed to: "Results showed that the simulations have underestimated observed surface aerosol levels, in particular black carbon (BC) and sulfate ($SO_4^{2-}$), by more than 50%, with the smallest biases calculated for the atmosphere-only simulations, where winds are nudged to reanalysis data." (Lines 31-33).*

• line 32: "The nudged simulations" have not been defined at this point. I recommend changing this to "...when winds were nudged to reanalysis data"
*Response: Done, see above response (Lines 33).*

• line 34: A change from "fully coupled simulations" to "simulations where atmosphere and ocean where coupled' or something similar might be better at this point.
*Response: We have now phrased the sentence as following: "In addition, simulations, where atmosphere and ocean are fully-coupled, had slightly smaller biases in aerosol levels compared to atmosphere only simulations without nudging." (Lines 35-37)*

• lines 37—48: None of the simulation names have been introduced at this point (naturally) and it might therefore be hard for the potential reader to grasp the general message of the abstract. I therefore recommend to re-write this paragraph. In my opinion a "maximum vs minimum effect"-type of discussion would be easier to digest at this point.
*Response: We have rewritten the paragraph accordingly (Lines 39-55).*

• line 46: remove "both"
*Response: Removed.*

• line 46—47: Change "In 2050" to "By 2050"?
*Response: Changed (Line 53).*

• line 52: "while scenarios no or little…" – add "with"
*Response: We have now rewritten this paragraph (Lines 57-60).*

• line 53: "lead" --> "leads"
*Response: See above response (Line 57).*

Introduction

• line 71: "This contribution ... puts" or "These contributions ... put" ?
*Response: Corrected accordingly (Line 77).*

• lines 80 --- 85: "BC" and "SO42-" have already been defined.
*Response: Corrected accordingly (Line 86-89).*

• line 90: I'm not sure myself: Is BC depositing on snow and ice or is BC being deposited, e.g. can you use the active form here?
*Response: Thanks for raising this. Indeed, the BC is deposited on snow. We have corrected this accordingly (Line 96).*

• line 93: While you talk about the lifetime and vertical extent effects here, if I understand the model description correctly, these effects are not included in your simulation, or are they?
*Response: These affects are taken into account in the model. Aerosols affect clouds via first indirect (CDNC) and semi-direct effects. We have now added the following text in lines 204-209: "The parameterization described by Menon and Rotstayn (2006) that we use only affects CDNC, not cloud droplet size, which is not explicitly calculated in GISS-E2.1. Following the change in CDNC, we do not stop the model from changing either LWP or precipitation rates, since the clouds code sees the different CDNC and responds accordingly. What we do not include is the 2nd indirect effect (autoconversion).".*

• lines 111—112: Is that global emissions?
*Response: "Global" is added (Line 125)*

Materials and Methods
• line 169—174: Can you elaborate on how that works? If Everything except dust and sea salt is externally mixed, does that mean that the model assumes separate sulphate, nitrate, BC and OC particles? How do you then treat the sulphate and nitrate coating of the dust particles?
*Response: Thanks for pointing this wrong phrase. We have now corrected this section (Lines 188-191).*

• Even though SOA production in the model is described in Tsigaridis and Kanakidou, maybe you could describe it briefly here as well. In particular, what are the assumptions of how SOA formation affects OC concentrations. This is important, as you attribute higher OC concentrations to higher SOA formation, but it is not clear, how that is modelled. Do you have separate SOA tracers or does VOC oxidation lead directly to OC production in the atmosphere? In the former case, how do you convert SOA into OC. Am I right in assuming that OC from the emission inventories is emitted as particulate matter?
*Response: We added the following in the text (Lines 196-202): "SOA is calculated from terpenes and other reactive volatile organic compounds (VOCs) using NOx-dependent calculations of the 2-product model, as described in Tsigaridis and Kanakidou (2007). Isoprene is explicitly used as a source, while terpenes and other reactive VOCs are lumped on a-pinene, taking into account their different reactivity against oxidation. The semi-volatile compounds formed can condense on all submicron particles except sea salt and dust. In the model, an OA to OC ratio of 1.4 used.". In addition, we now use OA instead of OC and explained this in lines 342-344 as: "The GISS-E2.1 ensemble has been evaluated against surface observations of BC, organic aerosols (sum of OC and secondary organic aerosols (SOA), referred as OA in the rest of the paper)…".*

• line 178—180: How does that work? If the model treats the first indirect (i.e. aerosol concentrations affecting CDNC and (I guess) cloud droplet size), how do you stop the model from changing LWP and precipitation rates?
*Response:We now added the following to the manuscript (Lines 204-209): "The parameterization described by Menon and Rotstayn (2006) that we use only affects CDNC, not cloud droplet size, which is not explicitly calculated in GISS-E2.1. Following the change in CDNC, we do not stop the model from changing either LWP or precipitation rates, since the clouds code sees the different CDNC and responds accordingly. What we do not include is the 2nd indirect effect (autoconversion)."*

• line 186: I guess this is also just the first indirect effect?

*Response: Yes. We have now clarified this (Line 215).*

• Section 2.2.3: Do I understand this correctly: Eclipse emissions have been complemented in some sectors by using CEDS emissions, while CEDS emissions are entirely "original", or did you also have to complement CEDS emissions in some sectors?

*Response: Yes this is correct. We have now slightly modified this section for clarity (Lines 282-290): "In the GISS-E2.1 Eclipse simulations, the non-methane volatile organic carbons (NMVOC) emissions are chemically speciated assuming the SSP2-4.5 VOC composition profiles. In the Eclipse simulations, biomass burning emissions are taken from the CMIP6 emissions, which have been pre-processed to include the agricultural waste burning emissions from the EclipseV6b dataset, while the rest of the biomass burning emissions are taken as the original CMIP6 biomass burning emissions. In addition to the biomass burning emissions, the aircraft emissions are also taken from the CMIP6 database to be used in the Eclipse simulations."*

• Lines 284 – 288: You have been quite thorough in explaining the differences between the ECLIPSE scenarios, but the differences between the different CEDS scenarios is quite compact. What, for instance, does "lowNTCF" mean?

*Scenario: We thank the reviewer to raise this. We have now described the different CMIP6 scenarios (Lines 316-331).*

• Section 2.2.3: How do the emissions and concentrations of Greenhouse gases evolve in the simulations? Are they kept fixed to capture the aerosol effect, or do they change? In the latter case, please elaborate on how you separate the aerosol effects from the Greenhouse gas effects.

*Response: All scenarios use the same prescribed global and annual mean GHG concentrations. We have now added this section to the manuscript (Lines 334-337): "We have employed prescribed global and annual mean greenhouse ($CO_2$ and $CH_4$) concentrations, where a linear increase in global mean temperature of 0.2 °C/decade from 2019 to 2050 was assumed, which are approximately in line with the simulated warming rates for the SSP2-4.5 scenario (AMAP, 2021).".*

• Section 2.2.3: If emissions are provided at 0.5x0.5° resolution, but the model operates at 2x2.5° resolution, I'm guessing you re-grid the emissions somehow?

*Response: Correct, we have now written this explicitly in the manuscript (Lines 281-282): "The Eclipse V6b and CEDS emissions on 0.5° × 0.5° spatial resolution are regridded to 2° × 2.5° resolution in order to be used in the various GISS-E2.1 simulations."*

• lines 317—326: As a side note, it has become more and more common to co-locate modelling data and observations in time to reduce the effects of observational "data sparseness" mentioned here. I understand that this is probably out of scope of this study, but worth considering in the future.

*Response: Thanks for the recommendations. We will consider this in future studies.*

Results

• Figure 3: It would be quite beneficial to add the station names to the figure. Especially because some of the stations are discussed in the text.

*Response:We have now added station numbers to Figure 3, instead of station names as this makes the plot very busy and messy. We have added the station numbers to the tables in the supplementary material.*

• lines 395-398: Could these high bias outliers be a problem with the representativeness of the observations (e.g. too few data points, or quickly changing orography)? Trapper Creek, for instance, is right next to another, blue, point.
*Response: Thanks for the suggestion. Yes, this can also be contributing to the biases due to large grid boxes in the model grid. We have also included this as a potential explanation to the bias (Lines 452-459): "Such underestimations at high latitudes have also been reported by many previous studies (e.g. Skeike et al., 2011; Eckhardt et al., 2015; Lund et al., 2017, 2018; Schacht et al., 2019; Turnock et al., 2020), pointing to a variety of reasons including uncertainties in emission inventories, errors in the wet and dry deposition schemes, the absence or underrepresentation of new aerosol formation processes, and the coarse resolution of global models leading to errors in emissions and simulated meteorology, as well as in representation of point observations in coarse model grid cells."*

• lines 433—436: Later in the article (line 774) you state that a higher cloud fraction may lead to higher in-cloud SO4 production – please add this statement also here.
*Response: We have now added this in lines 494-497: "The Eclipse_AMIP_NCEP simulation is biased higher (NMB=-53%) compared to the Eclipse_AMIP (NMB=-50%), probably due to higher cloud fraction simulated by the nudged version (see section 3.1.6), leading to higher in-cloud $SO_4^{2-}$ production."*

• Tables 3 and 4: Please consider breaking up these tables into two parts and displaying them in portrait mode. At least in electronic form it would make the manuscript easier to read.
*Response: We have now divided these tables into two (Tables 3a and 3b).*

• Why do the AMIP runs have such a high bias in SST, if SST is prescribed?
• lines 461—462: Is that due to model resolution? After all, SIC is prescribed, right?
*Response: This is due to differences with datasets used in the model input and the dataset used to evaluate the model. This is now described in lines 223-225 and 529-531.*

• lines 470—471: Do you mean the climatology of the cloud fraction for the entire year here?
*Response: Yes, we have now added this to the sentence (Line 546)*

• Figure 5a: I think here it would be worth mentioning that the seasonal trends in observed and modelled cloud fraction trends are reversed. Looking at panels b and c, it almost looks like the model produces to few water or mixed-phase clouds during the winter months, did I get this right?
*Response: We have now re-written this sentence (Lines 539-543): "All simulations overestimate the climatological (1995-2014) mean total cloud fraction by 21% to 25% during the extended winter months (October through February), where the simulated seasonality is anti-correlated in comparison to AVHRR CLARA-A2 observations, whereas, a good correlation is seen during the summer months irrespective of the observational data reference."*

• lines 474 – 478: This sentence is very hard to grasp: Less overestimation due to an underestimation? Do you mean to say that you trust AVHRR CLARA-A2 less than

CALIPSO, because CALIPSO does a better job at separating bright surfaces from clouds? Also, you could add in line 466 that there you compare to AVH*RR* data.

*Response: We have rephrased this sentence as (Lines 549-557): "The evaluation against CALIPSO data however shows much smaller biases (NMB = +3% to +6%). This is because in comparison to CALIPSO satellite that carries an active lidar instrument (CALIOP), the CLARA-A2 dataset has difficulties in separating cold and bright ice/snow surfaces from clouds thereby underestimating the cloudiness during Arctic winters. Here both datasets are used for the evaluation as they provide different observational perspectives and cover the typical range of uncertainty expected from the satellite observations. Furthermore, while the CLARA-A2 covers the entire evaluation period in current climate scenario, CALIPSO observations are based on 10-year data covering the 2007-2016 period."*

• Figure 6 and Section 6.2: I take it that by Arctic burden you mean the integral over all grid boxes between 60 and 90° north and over all vertical levels, but then using monthly averages? Why do you use the unit kTon in the text, but Tg in the figures?

*Response: This is correct, but the burdens are by default written as output so we do not do these calculations as a postprocessing of the data. We do not use monthly averages in these results in the manuscript. The figure is corrected now to included kTons instead of Tg.*

• line 533: What do you mean by "better resolved"?

*Response: We have now rewritten this section. We have now changed this sentence as (Lines 660-662): "This largest OA burden in the Eclipse_AMIP_NCEP simulation is attributed to the largest biogenic SOA burden calculated in this scenario, as well as a better-simulated transport from source regions due to the nudged winds (Figure S1).".*

• lines 541-542: If you term it "reduction", I guess the number should be positive...

*Response: We have now corrected this throughout the text.*

• line 549 and following: How has statistical significance been tested?

Response: We used Mann-Kendall trend analyses to calculate trends and statistical significance. We have now written this in Lines (626-627).

• line 554—555: See my comment in the Materials and Methods section. If OC is a separate tracer, you should explain somewhere, how a larger SOA production leads to larger OC concentrations. If it is what I think (i.e. you talk the sum of OC and SOA species), I suggest calling it something else. Maybe organic aerosol (OA) or organic matter (OM) would be suitable?

*Response: We now use OA instead of OC and explained this in lines 341-343 and 429-430 as: "The GISS-E2.1 ensemble has been evaluated against surface observations of BC, organic aerosols (sum of OC and secondary organic aerosols (SOA), referred as OA in the rest of the paper)...".*

• Figure S1: This links directly to the comment above. Without any explanation, it is not really understandable what you are showing here.

*Response: Corrected, see above response.*

• line 580: ...because CLE levels off earlier (no further legislation after this point?). The calculated trend cannot really be 2015—2050.

*Response: We cannot understand the comment. The CLE scenario is between 2015 and 2050.*

• Figure 7: What are you actually plotting here? From the explanation in the text (double call to the radiation code with and without aerosols) it sounds like you are showing the radiative effect due to aerosol-radiation interaction (REARI) (see Chapter 7.3.4.1 of the IPCC AR5), formerly termed the "direct radiative effect". A radiative forcing due to aerosol-radiation interaction (RFARI) would be the change in REARI relative to some reference point, e.g. preindustrial levels. Please elaborate.
*Response: This is correct. The plot and the text show the $RF_{ARI}$.*

• line 595 – 602: Why do you only talk about Eclipse here?
*Response: We thank the reviewer for pointing this out. We have now rewritten this part in a new separate sub-section (3.2.4).*

• 595 – 597: Why is that? This is quite a substantial difference – can this be explained by differences in aerosol burdens alone?
*Response: This difference is due to the larger sea-ice concentration simulated with the coupled model, leading to brighter surfaces compared to the AMIP simulations. This brighter surface also amplifies the effect of more positive BC forcing effect due to larger BC burdens simulated in the coupled model. This is now added in Lines 713-718.*

• line 601—602: What is the meaning of the third value here?
*Response: We have rewritten this section so this does not exist anymore. They used to show the mean of all eclipse future simulations (CLE and MFR combined), CLE simulations, and MFR simulations, respectively.*

• Figure 7: Why do the AMIP runs differ so much from the other simulations (2000—2015)? Also, there is visible difference between the black and brown lines (NINT_Cpl and CMIP6_Cpl_Hist?) in the anthropogenic aerosol radiative forcings, byt the same difference is not visible for the total aerosol radiative forcing – what is compensating for the difference here?
*Response: For the differences between AMIP and coupled simulations, please see the response above (Lines 718-718 in the manuscript). Regarding the difference in Figure 7 in the net vs anthropogenic $RF_{ARI}$ between the coupled NINT and the coupled OMA simulation is mainly driven by the dust and sea-salt $RF_{ARI}$. We have now explained this in the text (Lines 618-623).*

• If SOA can contribute to OC and if SOA can originate from both natural and anthropogenic sources, how can you separate the anthropogenic contribution of OC to the radiative forcing?
*Response: The model output includes speciated forcings for the anthropogenic, biomass burning, and SOA aerosols. We have now added the following sentence in the text (Lines 611-615): "The instantaneous forcings are calculated with a double call to the model's radiation code, with and without aerosols. The model outputs separate forcing diagnostics for anthropogenic and biomass burning BC and OC, as well as biogenic SOA, making it possible to attribute the forcing to individual aerosol species.".*

• Figure 8: how are the speciated forcings calculated?
*Response: GISS-E2.1 can calculate the RFARI by the double-call to the radiation code (Lines 611-615).*

• lines 640—651: This appears to be exactly the same text as lines 604—615.
*Response: We thank the reviewer for noticing this, we have now corrected this section.*

• line 650: What is higher to what here?
*Response: We have now rephrased this sentence (Lines 742-744): "Overall, the changes in the different aerosol species leads to a more negative aerosol forcing by mid-century (2030-2050) compared to the 1990-2010 period."*

• line 656: "sinnulations" --> "simulations"
*Response: Corrected.*

• line 657: You use the term "anomaly" the first time here – how is this calculated and what do you mean by "aerosol forcing anomaly"?
*Response: We have rewritten this section and this does not exist anymore.*

• Figure 9: In the figure you show only the surface temperatures between 2020 and 2050, but you talk a lot about temperature trends in earlier times – is there a reason for this? Also, it would be much easier to follow the discussion, if the observed trends would be added to the figure.
*Response: We have rewritten this part (Section 3.3.1). We have also added the observed values in the plot (now Figure 10).*

• Lines 665 – 673 I can't really believe the numbers you give in this paragraph. A 10° C/decade increase in surface air temperature is huge, even for the Arctic. As a reference, in the Figure 9 you show the surface temperatures between 2020 and 2050, which change by about 1-2°C in three decades. Please check your calculations or provide a figure, if the numbers are correct.
*Response: We thank the reviewer for pointing this error. We have now corrected this (lines 760-767).*

• 698: "warnings" --> "warmings"?
*Response: Corrected.*

• Figure 10: How statistically robust are these spatial distributions? Looking only at the SSP results (panels c, d and e), it looks like the changes are not very systematic in many regions, which makes me wonder how noisy the results are.
*We now plot the statistically-significant (student t-test) changes in the pots and it is highlighted in the text (Lines 778-779) and figure captions.*

• line 702: Figure 9 does not show SST.
*Response: Corrected.*

• line 712: Do you mean "Greenland sea"?
Response: Yes, this is now modified in the text (Lines 783-784).

• line 736: Here and in some other places where you compare the means of two time periods, you could consider replacing "... is projected to decrease by ... compared to ..." with "... is projected to be ... lower than ..."
*Response: We have modified the text as suggested by the reviewer.*

• Figure 11: Even though I the discussion is generally about the entire Arctic region, in this figure I'm wondering if it would be better to "zoom in" to where the changes are actually happening.
*Response: We have changed the figure as suggested by the reviewer (now Figure 12).*

• line 748: "Figure S1" --> "Figure S3"
• line 751: "Figure S2" shows SST
• line 754: "Figures S3—S7" --> "Figures S4—S7"
*Response: We have restructured these figures and removed S4-S7 as suggested by the other reviewer.*

Summary and Conclusions
• line 773: Like in the abstract, I would try to avoid using the names of the individual simulations in the conclusions.
*Response: Implemented the suggestion.*

• line 808: add "future"
*Response: Added.*

• lines 815 – 818. There appears to be one "Eclipse" too much.
*Response: Corrected.*

• line 826 – 829: Could one interpret this as the melting of sea ice acting buffering the changes in surface air temperatures?
*Response: This part has been changed and corrected (Lines 902-909).*

**Reviewer 2**

This study reports results from an extensive set of simulations with the GISS-E2.1 and two different emission inventories used to investigate the recent past and projected future changes in Arctic aerosols and aerosol-induced climate impacts. I find the study interesting and suitable within the scope of ACP. However, I also think it needs substantial further improvements before it can be accepted for publication. In particular, I find parts of the manuscript difficult to follow (the most notable example being the section on radiative forcing) and in some cases the possible reasons behind particular results could be better discussed. A better description of the experiments is needed for readers not within the AMAP group and I'm missing some context with impact due to other emissions than aerosols and precursors. In the introduction, the authors could better motivate why their study is important and timely. Finally, the figures could be visually more appealing. I think that improving the structure and readability should be quite feasible and some additional efforts will make a much stronger manuscript.

*Response: We thank the reviewer for the positive response to our manuscript. We have tried to implement the changes the reviewer has suggested here.*

Specific comments:

Line 30: "have been"? As in historical or in previous modeling work?

*Response: Modified the sentence as (Lines 31-33): "Results showed that the simulations have underestimated observed surface aerosol levels…".*

Line 33: Why also for climate parameters? What is different in the experimental setup?

*Response: we mean both concentrations and climate (meteorological) parameters were simulated better in the CMIP6 ensemble, the simulations are not different for the different parameters.*

Lines 37 onwards: would be useful to have the RF over the 1990-2014 period as well to understand changes in the scenarios.

*Response: We have rewritten this paragraph following suggestions from both reviewers (Lines 39-52).*

Line 46-48: Still due to changes in aerosols only? Should be more clear from the abstract hos greenhouse gases are treated.

*Response: Correct. We have now added the following sentence to the abstract (Lines 28-29), and in related section in the Materials and Methods; Lines 334-337): "… while global annual mean greenhouse gas concentrations were prescribed and kept fixed in all simulations.".*

Line 50-54: Similarly to the above comment, the role of aerosols vs. other emissions is a bit unclear.

*Response: Same as above response.*

Line 78: "mostly" – what are the remaining effects?

*Response: We have now extended this part as (Lines 84-85: "They mostly affect climate by altering the amount of solar energy absorbed by Earth, as well as changing the cloud properties and indirectly affecting the scattering of radiation,…".*

Line 88: "warming effects": here and in the following paragraphs I would suggest the authors be a bit more precise with regards to positive and negative RF versus warming/cooling, with the latter used only when actual temperature estimates are given. Furthermore, perhaps be clear whether it's surface warming or general.
*Response: We have tried to modify the overall text accordingly.*

Line 88: what are these aerosols? OC or all species?
*Response: We have added "organics" (Line 94).*

Line 90-104: the rapid adjustments from BC should be mentioned (ref to e.g. Stjern et al. 2017, Takemura 2019).

Stjern, C. W., Samset, B. H., Myhre, G., Forster, P. M., Hodnebrog, Ø. Andrews, T., … Voulgarakis, A. (2017). Rapid adjustments cause weak surface temperature response to increased black carbon concentrations. Journal of Geophysical Research: Atmospheres, 122, 11,462– 11,481. https://doi.org/10.1002/2017JD027326

Takemura, T., Suzuki, K. Weak global warming mitigation by reducing black carbon emissions. Sci Rep **9,** 4419 (2019). https://doi.org/10.1038/s41598-019-41181-6
*Response: We have now added a short section on the rapid adjustments (Lines 144-146).*

Line 109: "response through aerosols" – something strange with the language?
*Response: We have rephrased the sentence as (Lines 115-117): "The impact of aerosols on the Arctic climate change is mainly driven by a response to remote forcings (Gagné et al., 2015; Sand et al., 2015; Westervelt et al., 2015).".*

Line 109-onwards: Somewhere this section should mention/discuss long-range transport. While forcing exerted remotely is an important factor, there is also a lot of literature on the source attribution of Arctic aerosols. Given that Arctic burdens are shown later, the LRT is relevant to understand to interpret changes in burden over time.
*Response: We have now added a short section on long-range transport (Lines 117-124).*

Line 111: is this per unit global sulfur emission?
*Response: Global, added to the sentence (Line 125).*

Line 131: I think this paper actually removed aerosols entirely? Relevant for the response.
*Response: Correct, we have now added this to the sentence (Lines 146-149): "Samset et al. (2018), using a multi-model ensemble of ocean coupled Earth system models (ESMs), where aerosol emissions were either kept at present-day conditions, or anthropogenic emissions of $SO_2$, and fossil fuel BC and OC were set to zero, showed…".*

Section 2.2: perhaps reconsider the number of small paragraphs? It becomes a bit broken up and the first sentences of the section are repeated later.
*Response: We have slightly modified the section accordingly.*

Section: 2.3: this is probably clear to people who are familiar with the AMAP runs, but to me it's very unclear how other emissions (CO2, etc.) are treated in these experiments. Which in turn makes results hard to interpret. I think experiments could be a bit better explained.
*Response: We have now provided with explanations on the different emission scenarios (in lines 315-331 ) and AMAP experiments (Lines 334-337).*

Line 303: when I think of IMPROVE, I don't exactly think of the Arctic. Perhaps it could be useful to give the number of stations in each network that are within the relevant region? (yes, there are SI tables, but to help the reader).
*Response: The IMPROVE measurements that are in the Arctic (>60oN), are all in Alaska. Thus, we have changed the text to make that clear (replacing "United States" with "Alaska"). There were 5 measurement locations in Alaska, all associated with IMPROVE, but some obtained from their PIs since they were difficult to obtain from the general IMPROVE data portal. There were 6 measurement locations in Europe, though not all associated with the EMEP network, and 1 measurement location in Canada/CABM. We don't include the number of sites in the text next to the networks however, since it is somewhat complicated when obtaining measurements from individual PIs instead of the network portals, and we wouldn't want to be in error.*

Section 2.4.1: In later tables and figures satellite observations of AOD are mentioned, but I can't see those described here? Please clarify.
*Response: In this paper we used a combined product developed by Sogacheva et al. (2020) by merging AOD from various different satellite products. We did not use here individual satellite AOD products. There is very detailed information in Sogacheva et al. (2020) about how they do it, and it is very technical. We think it would be unnecessary to discuss the details of that here.*

Line 383-384: I don't understand this sentence and relationship. Please consider rephrasing.
*Response: We have rephrased this sentence as (Lines 434-436): "The monthly observed and simulated time series for each station are accumulated per species in order to get a full Arctic timeseries data, which also includes spatial variation, to be used for the evaluation of the model."*

Section 3.1: In general, an indication of the interannual variation around the climatological mean would be very useful, at least for observations when this can be added to the figures.
*Response: We have now updated the Figures 2,4, and 5 to include the interannual variation in the observations and simulations.*

Section 3.1.1: perhaps discuss the seasonal differences in the underestimation better.
*Response: We have now added some more explanation throughout the model evaluation section.*

Moreover, I'm not convinced by the inclusion of individual ensemble members as separate experiments. I think it rather adds unnecessary complexity and, in addition given how briefly these results are discussed in the text, could rather be an average and a ± range. (This goes for climate variables as well.)
*Response: We now present only the ensemble means of the individual experiments.*

Line 423-430: From Figure 3, it seems that OC is very well captured. This seems worth describing and explaining. It surprises me that the seasonal cycle of the observations is so different from BC. Is it a dominance by biogenic SOA?
*Response: We have now added the following explanation (Lines 485-491): "As can be seen in Figure S1, the OA levels are dominated by the biogenic SOA, compared to anthropogenic and biomass burning OA. While OC and BC are emitted almost from similar sources, this biogenic-dominated OA seasonality also explains why simulated BC seasonality is not as*

*well captured, suggesting the underestimations in the anthropogenic emissions of these species, in particular during the winter. "*

Figure 2: Is this an average over all stations? Please be specific.
*Response: As explained in lines 434-436, the figure shows the monthly observed and simulated time series for each station are accumulated per species in order to get a full Arctic timeseries data.*

Figure 1: Needs improvement. Difficult to see different colors.
*Response: We have tried to increase visibility in the figure.*

Section 3.1.2: Is it possible to place the GISS model in context of other CMIP model's performance here? Is this a typical feature? Would be useful.
*Response: There is already some discussion on this in later sections (3.3.1 and 3.3.2) for surface temperatures and sea-ice extent, however focusing on the projected changes rather than model evaluation. Comparing GISS-E2.1 with other CMIP6 models is out of the scope of this paper and requires large amount of analysis.*

Lines 555 onwards: I find the discussion around the role of SOA hard to follow. So OC in figure 6 includes SOA? What is the OA-OC conversion factor? Furthermore, I think more explanation of why these differences exist is needed, rather than just attributing one to the other.
*Response: We now use OA instead of OC, which is the sum of OC and SOA (Lines 341-343).*

Section 3.2: use same unit as figure 6?
*Response: We have changed figure unit to kTon to be consistent with the text.*

Section 3.3: this section needs some improvement.
- I would recommend using terminology RFari and RFaci. I also don't think you need to keep saying TOA radiative forcing, that's in the definition. Will help improve readability as well.

*Response: We now use $RF_{ARI}$ throughout the text.*

- The section is difficult to follow with the many different time periods used. For instance, lines 595-602 gives a set of numbers that are not quite different from the sum of the aerosol RF in Figure 8. Lines 595-602 seems to give the RF due to changes in aerosols from 1990 to 2010 and then from 2030 to 2050, but what but the RF due to the difference from 2010 – 2030 and 2050?

*Response: We have now restructured this section by combining with the burdens for the individual aerosol species, provided extra tables and tried to focus on the differences in the text.*

- Figure 8: RF is already a delta, a perturbation vs. a baseline, so it's not clear to me what this figure is showing.

*Response: The figure shows the difference between the 2030-2050 mean and 1990-2010 RFARI values for the different aerosol species.*

- Some RF numbers have a ± range, but it doesn't seem to be the case for the numbers in table 4.

*Response: We now present the mean values in the text.*

- Line 654: here the 2015-2050 forcing is also introduced. To me, this is a more relevant measure than the e.g. the forcing in 2050 relative to 2030 because in many cases, the emission changes are not that large from 2030 to 2050. At the very least, give this period in table 4 and hint to the reader at the beginning of the section that it will be mentioned. And why calculate this relative to 2015 and not 2010?

*Response: We now present the results from the difference of 1990-2010 and the 2030-2050 means to focus more on the difference between the future and the past.*

- At the beginning of the paper, you talk about how Arctic climate change is primarily due to remote forcing. For this reason, I think it would be useful to give the reader an idea of also the global mean RF. This can be done in the SI, but would also enable comparison with previous work.

*Response: We now have a new section (Section 3.2.4) that focuses on the net aerosol forcing, where we also present briefly the global mean RF$_{ARI}$. We also present the global spatial distribution of the difference between the 1990-2010 and 2030-2050 mean RF$_{ARI}$ in Figure S2 (lines 750-753).*

- One or two figures of the geographical distribution of forcing would also be useful. Can be sub-panels of figure 7.

*Response: We have added a new figure (Figure 8) showing the spatial distribution of the Arctic difference between the 1990-2010 and 2030-2050 mean RF$_{ARI}$*

Line 667-671: the model gives a 10 degree per decade change compared to 2 degrees from the observations? That seems like a very noticeable difference that I don't think you can just mention briefly like this, but needs more attention. What does this imply for confidence in any of the projections?
*Response: We thank the reviewer for noticing this error. We have now corrected this (Lines 758-767).*

Line 698: Not sure lowNTCF has been defined anywhere?
*Response: We have now defined all the different CMIP6 scenarios in section 2.3 (Lines 316-331).*

Line 754-755: Here you have 3 big figures in the SI that hardly show anything but white map and then show that anything they do show is not really significant. I would perhaps reconsider the usefulness and need for these figures.
*Response: We agree with the reviewer. We have removed these figures.*

Figure S8: I'm not sure it's correct to refer to projected changes as anomalies? And, is the isoprene plot referred to anywhere in the text? This is important for the discussion about SOA burden.
*Response: We have rewritten this part and now we do not use this term anymore.*

Line 765: 1990-2014? But RF was discussed based on 1990-2010? Please clarify.
*Response: This only defines the simulation period, which is from 1990 to 2050. The analyses are conducted in the manuscript focuses on the differences between the 1990-2010 and 2030-2050 periods.*

---

## Author Response (AR2)

Response to the editor

We would like to thank the editor for the comments and suggestions. We have now modified the manuscript based on these comments and hope that this new version is now suitable for publication.

1) As both reviewers were confused by the terms being used to describe aerosol direct and indirect effects, the former has been changed to radiative forcing due to aerosol-radiation interactions (RFari). It needs more clarification. Based on my understanding, the difference in TOA fluxes between the double calls to radiation, with and without certain aerosol species, is not strictly the "radiative forcing" used in IPCC AR5, which is defined as the TOA flux changes in a given time period (e.g., 2000 to 2010) relative to a reference state (e.g., in 1750 or 1850). There is no problem to define the terms in a different way, but the concern is about the confusion caused by comparing the forcing magnitude to others in the literature. I suggest making this explicit in the manuscript, especially in the abstract and summary section where there is not enough context for the numbers. I have also seen in other studies that this term is called "radiative effect due to aerosol-radiation interactions (REari). This could be an alternative to distinguish from the RFari used in AR5 and many other studies. Please revise the manuscript at your own discretion.

*Response: We use instantaneous direct aerosol forcing numbers RFari using the diagnostics of the historical transient simulations, based on double calls to the radiation including and excluding aerosol forcing effects, as explained in Bauer et al. (2020). We have now modified the text as (Lines 616-622): " RFARI is calculated  as the sum of shortwave and longwave forcing from the individual aerosol species between 1850 and 2050 are presented in Figure 7. It is important to note that the present study uses the instantaneous forcing diagnostics from the model, which are calculated with a double call to the model's radiation code, with and without aerosols, as described in Bauer et al. (2020) and Miller et al. (2021), and not the effective radiative forcing. The transient cloud radiative effect in GISS-E.2.1 follows Ghan (2013), which calculates the difference in cloud radiative forcing with aerosol scattering and absorption omitted (Bauer et al., 2020). However, the present study only focuses on the RFARI."*

2) Line 135: typo for "warming"

*Response: Corrected (Line 135).*

3) Line 306-307: Are the winds nudged all the way from the surface to the model top? Usually, wind nudging starts from a certain height near the top of boundary layer to avoid undesired impact on the calculation of surface fluxes and emissions of natural aerosols (e.g., sea salt and dust). Please clarify.

*Response: The winds are nudged starting from the first model layer and the dust concentrations are tuned to match the observed dust AOD. We have now added this to the manuscript (Lines 309-310): "The nudging extents from the first model layer up to 10 hPa, which is the top of the NCEP input.", and (Lines 221-222): "Dust concentrations are tuned to match the observed dust aerosol optical depth (AOD).".*

4) Line 520-523: I think this statement is an over-interpretation of the results and could be very misleading. This is also in contradiction to results from other models in the literature. Aerosol indirect effects in the current model (GISS-E2.1) are incomplete. There are also large uncertainties in the model simulated Arctic local aerosols, as well as clouds, and likely in the mid-latitudes. Please tune down the claim and/or provide the right context and preassumption.

*Response: We have now changed the sentence as (Lines 525-528): "This, together with the very low biases in surface temperatures suggests that the effects of the anthropogenic aerosols on the Arctic climate via radiation is not the main driver in comparison to cloud indirect effects and forcing from greenhouse gases."*

5) Line 528-529: As brought up by one of the reviewers, it's confusing to say, "the atmosphere-only runs underestimated SSTs…" since the AMIP-type runs do not compute SSTs. If you treat the SST data for model evaluation as truth, then the statement should be something like "SSTs input provided to the atmosphere-only runs has a bias of …". The natural question from reader would be "Why are the biased SSTs used to drive the AMIP-type runs in the first place?". More clarification in addition to citation of two references would be helpful.

*Response: We have changed this sentence as (Lines 534-537): "The negative bias in atmosphere-only simulations is due to the different datasets used to drive the model, which a combined product of HadISST and NOAA-OI2 (Reynolds et al., 2002) and to evaluate the model (Rayner et al., 2003), which is only HadISST."*

6) Line 584-586: Please provide a reference for this statement. My understanding is that the transport pathways to the Arctic depend on aerosol source origins.

*Response: We have now provided these references (Lines 589-591): "Furthermore, the aerosol and pollution transport into the Arctic typically occurs in the lowermost troposphere where liquid water clouds are prevalent during late spring and summer seasons (Stohl, 2006; Law et al., 2014; Thomas et al., 2019)."*

7) Line 858-860: Aren't the model simulations showing an overestimation of Arctic cloud fraction and LWP most of the time (Figure 5)? Moreover, I don't think it's appropriate to attribute the bias in clouds primarily to the availability of CCN unless further evidence is provided from the simulations.

*Response: We thank the editor for raising this wrong statement. We now changed it as (Lines 870-874): "
[revised manuscript text omitted]